# Limits, approximation and size transferability for GNNs on sparse graphs via graphops

**Thien Le**
MIT
thienle@mit.edu

**Stefanie Jegelka**
TU Munich and MIT
stefje@csail.mit.edu

## Abstract

Can graph neural networks generalize to graphs that are different from the graphs they were trained on, e.g., in size? In this work, we study this question from a theoretical perspective. While recent work established such transferability and approximation results via graph limits, e.g., via graphons, these only apply nontrivially to dense graphs. To include frequently encountered sparse graphs such as bounded-degree or power law graphs, we take a perspective of taking limits of operators derived from graphs, such as the aggregation operation that makes up GNNs. This leads to the recently introduced limit notion of graphops (Backhausz and Szegedy, 2022). We demonstrate how the operator perspective allows us to develop quantitative bounds on the distance between a finite GNN and its limit on an infinite graph, as well as the distance between the GNN on graphs of different sizes that share structural properties, under a regularity assumption verified for various graph sequences. Our results hold for dense and sparse graphs, and various notions of graph limits.

## 1 Introduction

Since the advent of graph neural networks (GNNs), deep learning has become one of the most promising tools to address graph-based tasks (Gilmer et al., 2017; Scarselli et al., 2009; Kipf and Welling, 2017; Bronstein et al., 2017). Following the mounting success of applied GNN research, theoretical analyses follow with many works studying GNNs' representational power (Azizian and Lelarge, 2021; Morris et al., 2019; Xu et al., 2019, 2020; Garg et al., 2020; Chen et al., 2020; Maron et al., 2019; Loukas, 2020a,b; Abboud et al., 2021).

A hitherto less addressed question of practical importance is the possibility of size generalization, i.e., transferring a learned GNN to graphs of different sizes (Ruiz et al., 2023a; Levie et al., 2022; Xu et al., 2021; Yehudai et al., 2021; Bevilacqua et al., 2021; Chuang and Jegelka, 2022; Roddenberry et al., 2022; Maskey et al., 2022, 2023), especially for sparse graphs. For instance, it would be computationally desirable to train a GNN on small graphs and apply it to large graphs. This question is also important to judge the reliability of the learned model on different test graphs. To answer the size generalization question, we need to understand under which conditions such transferability is possible – since it may not always be possible (Xu et al., 2021; Yehudai et al., 2021; Jegelka, 2022) – and what output perturbations we may expect. For a formal analysis of perturbations and conditions, we need a suitable graph representation that captures inductive biases and allows us to compare models for graphs of different sizes. *Graph limits* can help to formalize this, as they help understand biases as the graph size tends to infinity.

Formally, *approximation theory* asks for bounds between a GNN on a finite graph and its infinite counterpart, while *transferability* compares model outputs on graphs of different sizes.

37th Conference on Neural Information Processing Systems (NeurIPS 2023).

The quality of the bounds depends on how the two GNNs (and corresponding graphs) are intrinsically linked, in particular, to what extent they share relevant structure. This yields conditions for size generalization. For example, the graphs could be sampled from the same graph limit (Ruiz et al., 2023a) or from the same random graph model (Keriven et al., 2020).

In particular, Ruiz et al. (2023a) study approximation and transferability via the lens of *graphons* (Lovász, 2012; Lovász and Szegedy, 2006), which characterize the limits of *dense* graphs. Yet, many real-world graphs are not dense, for instance, planar traffic networks, power law graphs, polymer graphs, Hamming graphs (including hypercubes for error-correcting code), or grid-like graphs e.g., for images. For *sparser* graphs, the correct notion of limit suitable for deep learning is still an open problem, as typical bounded-degree graph limits such as the Benjamini-Schramm limit of random rooted graphs (Benjamini and Schramm, 2001), or graphings (Lovász, 2012) are less well understood and often exhibit pathological behaviors (see Section 2.1). Limits of intermediate graphs, such as the hypercubes, are even more obscure. Hence, understanding limits, inductive biases and transferability of GNNs for sparse graphs remains an open problem in understanding graph representation learning.

This question is the focus of this work. To obtain suitable graph limits for sparse graphs and to be able to compare GNNs on graphs of different sizes while circumventing challenges of sparse graph limits, we view a graph as an *operator* derived from it. This viewpoint is naturally compatible with GNNs, as they are built from convolution/aggregation operations. We show how the operator perspective allows us to define limits of GNNs of infinite sequences of graphs. We achieve this by exploiting the recently defined notion of *graphop*, which generalizes graph shift operators, and the *action convergence* defined in the space of graphops (Backhausz and Szegedy, 2022). Our definition of GNN limits enables us to prove rigorous bounds for approximation and transferability of GNNs for sparse graphs. Since graphops encompass both graphons and graphings, we generalize similar bounds for graphon neural networks (Ruiz et al., 2023a; Maskey et al., 2023) to a much wider set of graphs.

Yet, using graphops requires technical work. For instance, we need to introduce an appropriate discretization of a graphop to obtain its corresponding finite graph shift operators. We use these operators to define a generalized graphop neural network that acts as a limit object, with discretizations that become finite GNNs. Then we prove approximation and transferability results for both the operators (graphops and their discretizations) and GNNs.

**Contributions.** To the best of our knowledge, this is the first paper to provide approximation and transferability theorems specifically for sparse graph limits. Our main tool, graphops, has not been used to study GNNs before, although viewing graphs as operators is a classic theme in the literature. Our specific contributions are as follows:

1. We define a *graphop convolution*, i.e., an operator that includes both finite graph convolutions and a limit version that allows us to define a limit object for GNNs applied to graphs of size $n \to \infty$.

2. We rigorously prove an approximation theorem (Theorem 2) that bounds a distance between a graphop $A$ (acting on infinite-dimensional space) and and appropriate discretization $A_n$ (acting on $\mathbb{R}^n$), in the $d_M$ metric introduced by Backhausz and Szegedy (2022). Our result applies to a more general set of nonlinear operators, and implies a transferability bound between finite graphs (discretizations) of different sizes.

3. For neural networks, we present a quantitative approximation and transferability bound that guarantees outputs of graphop neural networks are close to those of the corresponding GNNs (obtained from discretization).

## 1.1 Related work

A summary of comparisons between our framework and related papers is in Table 1.

In structure, the closest related work is (Ruiz et al., 2023a), which derives approximation and transferability theorems for *graphon* neural networks, i.e., *dense* graphs. For graphons, the convolution kernel has a nice spectral decomposition, which is exploited by Ruiz et al.

---

[1]Rate of convergence to a small positive constant. For convergence to 0, Ruiz et al. (2020) showed the rate of $O(n^{-1/2})$.

| | Sparse | | Dense |
|---|---|---|---|
| | Bounded-degree | Relatively-sparse | |
| Number of edges | $\Theta(n)$ | $\Theta(n\log n)$ | $\Theta(n^2)$ |
| Examples covered under our assumptions | infinite grids, polymer graphs | hypercubes, Hamming graphs | graphons |
| Graphons (Ruiz et al., 2023a) | | | $O(n^{-1})$[1] |
| Unbounded graphons (Maskey et al., 2023) | | | inexplicit |
| Random graph model (Keriven et al., 2020) | | $O((\log n)^{-1/2})$ | $O(n^{-1/2})$ |
| Spectral methods (1 layer) (Levie et al., 2022) | | inexplicit | inexplicit |
| Graphings (1 layer) (Roddenberry et al., 2022) | inexplicit | | |
| $P$-operators and graphops (ours) | $O(n^{-1/2})$ | $O(n^{-1/2})$ | $O(n^{-1/2})$ |

Table 1: Summary of our results compared to related work. Quantitative results (e.g. $O(n^{-1/2})$) upper-bound the distance between GNNs on sampled graphs of size $n$ and the limiting object in term of $n$ (in an appropriate metric and limit notion). Empty cells are graph models where the approaches in the corresponding papers do not apply to or give trivial bounds (e.g. bounds that compare to a constant-0 graphon). "Inexplicit" refers to asymptotic results where rates of convergence is not explicit.

(2023a). In contrast, *sparse* graph limits are not known to enjoy nice convergence of the spectrum (Backhausz and Szegedy, 2022; Aldous and Lyons, 2007), also Appendix C.1, so we need to use different techniques. Since the notion of graphop generalizes both dense graph limits and certain sparse graph limits, our results apply to dense graphs as well. Our assumptions and settings are slightly different from Ruiz et al. (2023a). For instance, they allow the convolution degree $K \to \infty$ and perform the analysis in the spectral domain, whereas our $K$ is assumed to be a fixed finite constant. As a result, their bound has better dependence of $O(1/n)$ on $n$–the resolution of discretization, but does not go to 0 as $n \to \infty$. Ours have extra dependence on $K$ and a slower rate of $O(n^{-1/2})$ but our bounds go to 0 as $n \to \infty$. Maskey et al. (2023) obtains further results for graphons on unbounded domain. Ruiz et al. (2023b) studies the spectrum of sparser (but still $\Theta(n^2)$ edges) graphons.

Other works use other notions than graph limits to obtain structural coherence. Levie et al. (2022) obtain a transferability result for spectral graph convolution networks via analysis in frequency domains. They sample finite graphs from general topologies as opposed to a graph limit. Their graph signals are *assumed* to have finite bandwidth while ours is only assumed to be in $L^2$. Their signal discretization scheme is assumed to be close to the continuous signals, while ours is proven to be so. Roddenberry et al. (2022) address sparse graphs and give a transferability bound between the loss functions of two random rooted graphs. However, the metric under which they derive their result is rather simple: if the two graphs are not isomorphic then their distance is constant, otherwise, they use the Euclidean metric between the two graph signals. This metric hence does not capture combinatorial, structural differences of functions on non-isomorphic graphs. To study transferability, Keriven et al. (2020) sample from standard random graph models, resulting in a bound of order $O(n^{-1/2})$ for dense graph and $O((\log n)^{-1/2})$ for relatively-sparse graphs. They do not cover bounded-degree graphs and their bounds hold with high probability. In general, having access to deterministic graph limit is considered a weaker assumption than having access to truly random graphs since large graphs satisfy some notion of 'almost randomness' (e.g. via Szemerédi regularity lemma). There are fascinating, tangile separations between graph limits and random graphs, especially among sparse graphs, that are outside the scope of this paper.

Adjacent to transferability studies, Bevilacqua et al. (2021) (also inspired by Lovász and Szegedy (2006)) uses induced homomorphism density to construct a graph representation that works across graph sizes and demonstrates empirical gains in using this larger representation. Maskey et al. (2022) uses graph limits to deduce generalization bounds for GNNs.

## 2 Background

**Notation** Let $\mathbb{N}$ be $\{1, 2, \ldots\}$ and write $[n] = \{1, \ldots, n\}$ for any $n \in \mathbb{N}$. For a scalar $\alpha \in \mathbb{R}$ and a set $S \subset \mathbb{R}$, let $\alpha S = \{\alpha s : s \in S\}$. 'A.e.' stands for 'almost everywhere'.

For a measure space $(\Omega, \mathcal{B}, \mu)$ and $p \in [1, \infty]$, denote by $L^p(\Omega)$ the corresponding $L^p$ function spaces with norm $\|\cdot\|_p : f \mapsto (\int_\Omega |f|^p d\mu)^{1/p}$. For any $p, q \in [1, \infty]$, define the operator norms $\|\cdot\|_{p \to q} : A \mapsto \sup_{v \in L^\infty} \|vA\|_q / \|v\|_p$.

For function spaces, we use $\mathcal{F} = L^2([0,1])$ and $\mathcal{F}_n = L^2([n]/n)$, for any $n \in \mathbb{N}$. For any $L^p$ space $\mathcal{H}$, denote by $\mathcal{H}_{[-1,1]}$ the restriction to functions with range in $[-1,1]$ a.e. and $\mathcal{H}_{\mathrm{Lip}(L)}$ the restriction to functions that are $L$-Lipschitz a.e. and $\mathcal{H}_{\mathrm{reg}(L)} = \mathcal{H}_{[-1,1]} \cap \mathcal{H}_{\mathrm{Lip}(L)}$.

**Graph neural networks (GNNs)** GNNs[2] are functions that use graph convolutions to incorporate graph structure into neural network architectures. Given a finite graph $G = (V, E)$ and a function $X : V \to \mathbb{R}$ (called *graph signal* or *node features*), a GNN $\Phi_F$ ($F$ for 'finite') with $L$ layers, $n_i$ neurons at the $i$-th layer, nonlinearity $\rho$ and learnable parameters $h$, is:

$$\Phi_F(h, G, X) = X_L(h, G, X), \tag{1}$$

$$[X_l(h, G, X)]_f = \rho\left( \sum_{g=1}^{n_{l-1}} A_{l,f,g}(h, G)[X_{l-1}]_g \right), \qquad l \in [L], f \in [n_l] \tag{2}$$

$$X_0(h, G, X) = X, \tag{3}$$

where $[X_l]_f$ is the output of the $f$-th neuron in the $l$-th layer, which is another graph signal. The input graph information is captured through order $K$ *graph convolutions* $A_{l,f,g}(h, G) := \sum_{k=0}^{K} h_{l,f,g,k} GSO(G)^k$, where $GSO(G)$ is a *graph shift operator* corresponding to $G$ — popular examples include the adjacency matrix or the Laplacian (Kipf and Welling, 2017; Levie et al., 2022). The power notation is the usual matrix power, while the notation $h_{l,f,g,k}$ highlights that there is a learnable parameter for each convolution order $k$, between each neuron $f$ and $g$ from layer $l-1$ to layer $l$ of the neural network. Thus, the number of learnable parameters in a GNN does not depend on the number of vertices of the graph.

## 2.1 Graph limits

Graph limit theory involves embedding discrete graphs into rich topological or geometric spaces and studying the behavior of convergent (e.g. in size) graph sequences.

**Dense graphs** A popular example of graph limits are graphons - symmetric $L^1([0,1]^2)$ (Lebesgue-measurable) functions whose value at $(x, y)$ can be thought of (intuitively) as the weight of the $xy$-edge in a graph with vertices in $[0,1]$. Convergence in this space, under the *cut metric* (see Appendix A for the exact definition), is dubbed *dense graph convergence* because for any $W \in L^1([0,1]^2), \|W\|_\square = 0$ iff $W = 0$ outside a set of Lebesgue measure 0. This implies that graphs with a subquadratic number of edges, such as grids or hypercubes, are identified with the empty graph in the cut norm. Dense graph convergence is very well understood theoretically and is the basis for recent work on GNN limits (Ruiz et al., 2023a).

**Sparse graphs** *Graphing* (Lovász, 2012), is a direct counterpart of a graphon for sparse graphs. Recall that graphons are not suitable for sparse graphs because the Lebesgue measure on $L^2([0,1]^2)$ is not fine enough to detect edges of bounded-degree graphs. Therefore, one solution is to consider other measure spaces. Graphings are quadruples $(V, \mathcal{A}, \lambda, E)$ where $V$ and $E$ are interpreted as the usual vertex and edge sets and $(V, \mathcal{A}, \lambda)$ together form a Borel measure such that $E$ is in $\mathcal{A} \times \mathcal{A}$ satisfying a symmetry condition. While Lebesgue measures are constructed from a specific topology of open sets on $\mathbb{R}$, for graphings, we are allowed the freedom to choose a different topological structure (for instance a *local topology*) on $V$. The definition of graphings is theoretically elegant but harder to work with since the topological structures are stored in the $\sigma$-algebra.

Furthermore, a famous open conjecture by Aldous and Lyons (2007) asks whether all graphings are weak local limits of some sequence of bounded-degree graphs. The unresolved conjecture of Aldous and Lyons means that one cannot simply take an arbitrary graphing and be guaranteed a finite bounded-degree graph sequence converging to said graphing, which is the main approach in Ruiz et al. (2023a) for dense graphs. As a result, we expect

---

[2]The architecture is known as graph convolutional networks, but we call them GNNs for brevity.

some regularity assumptions on the graph sequence for any work that handles sparse graph limits, unless the authors try to tackle this conjecture itself. A self-contained summary of graphings within the scope of this paper is provided in Appendix C. Infinite paths and cycles also have nice descriptions in terms of graphings (also in Appendix C), which we will use in our constructions for Lemma 2.

## 2.2 Graphops and comparing across graph sizes

More recently, Backhausz and Szegedy (2022) approach graph limits from the viewpoint of limits of operators, called *graphops*. This viewpoint is straightforward for finite graphs: both the adjacency matrix and Laplacian, each defining a unique graph, are linear operators on $\mathbb{R}^{\#\text{vertices}}$. Moreover, viewing graphs as operators is exactly what we do with GSOs and graph convolutions. Hence, graphop seems to be an appropriate tool to study GNN approximation and transferability. On the other hand, there are challenges with this approach: being related to graphings, they inherit some of graphings' limitations, such as the conjecture of Aldous and Lyons (2007) and discontinuity of eigenvalues at the limit (Appendix C.1). Moreover, to understand GNN transferability from size $m$ to $n$, one needs to compare an $m \times m$ matrix with an $n \times n$ matrix, which is nontrivial. This is done by comparing their actions on $\mathbb{R}^m$ versus $\mathbb{R}^n$. It turns out that these actions, under an appropriate metric, define a special mode of operator convergence called *action convergence*. The resulting limit objects are well-defined and nontrivial for sparse graphs and intermediate graphs, while also generalizing dense graphs limits. We will describe this mode of convergence, the corresponding metric, and our own relaxation of it later in this section.

We now describe how graphs of different sizes can be compared through the actions of their corresponding operators on some function spaces.

**Nonlinear (not necessarily linear) $P$-operators** For an $n$-vertex graph, its adjacency matrix, Laplacian, or random walks kernels are examples of operators on $L^p([n]/n)$. To formally generalize to the infinite-vertex case, Backhausz and Szegedy (2022) use $P$-*operators*, which are linear operators from $L^\infty(\Omega)$ to $L^1(\Omega)$ with finite $\|A\|_{\infty \to 1}$. In this paper, we further assume they have finite $\|\cdot\|_{2\to2}$ norm but are not necessarily linear. This allows us to consider nonlinear GNN layers and the whole GNN itself in the same framework.

**Graphops** $P$-operators lead to a notion of graph limit that applies to both dense and sparse graphs. *Graphops* (Backhausz and Szegedy, 2022) are positivity-preserving[3], self-adjoint $P$-operators. Adjacency matrices of finite graphs, graphons (Lovász and Szegedy, 2006), and graphings (Lovász, 2012) are all examples of graphops.

**$(k, L)$-profile of a nonlinear $P$-operator** Actions of graphops are formally captured through their $(k, L)$-profiles, and these will be useful to compare different graphops. Pick $k \in \mathbb{N}$, $L \in [0, \infty]$ and $A$ a $P$-operator on $(\Omega, \mathcal{B}, \mu)$. Intuitively, we will take $k$ samples from the space our operators act on, apply our operator to get $k$ images, and concatenate samples and images into a joint distribution on $\mathbb{R}^{2k}$, which gives us one element of the profile. For instance, for $n$-vertex graphs, the concatenation results in a matrix $M \in \mathbb{R}^{n \times 2k}$, so each joint distribution is a sum (over rows of $M$) of $n$ Dirac distributions. In the limit, the number of atoms in each element of the profile increases, and the measure converges (weakly) to one with density. More formally, denote by $\mathcal{D}(v_1, \ldots, v_k)$ the pushforward of $\mu$ via $x \mapsto (v_1(x), \ldots, v_k(x))$ for any tuple $(v_i)_{i \in [k]} \in L^2(\Omega)$. The $(k, L)$-*profile* of $A$ is:

$$\mathcal{S}_{k,L}(A) := \{\mathcal{D}(v_1, \ldots, v_k, Av_1, \ldots, Av_k) : v_i \in L^\infty_{\text{reg}(L)}(\Omega), i = 1 \ldots k\}. \tag{4}$$

Formally, denote by $\mathcal{P}(\mathbb{R}^k)$ the set of Borel probability distributions over $\mathbb{R}^{2k}$. Regardless of the initial graph size, or the space on which the operators act, $(k, L)$-profiles of $A$ are always some subsets of $\mathcal{P}(\mathbb{R}^{2k})$ which allow us to compare operators acting on different spaces.

**Convergence of $P$-operators** We compare two profiles (closed subsets $X, Y \subset \mathcal{P}(\mathbb{R}^{2k})$) via a Hausdorff metric $d_H(X, Y) := \max(\sup_{x \in X} \inf_{y \in Y} d_{LP}(x, y), \sup_{y \in Y} \inf_{x \in X} d_{LP}(x, y))$.

---

[3]action on positive functions results in positive functions. This condition can be swapped out for 'positiveness' ($\langle Ax, x \rangle > 0, \forall x \in \text{Dom}(A) \backslash \{0\}$) to allow for Laplacians.

Here, $d_{LP}$ is the Lévy-Prokhorov metric on $\mathcal{P}(\mathbb{R}^{2k})$ (see exact definition in Appendix A), which metrizes weak convergence of Borel probability measures, and translates action convergence to weak convergence of measures. Finally, given any two $P$-operators $A, B$, we can compare their profiles across all different $k$ at the same time as

$$d_M(A, B) := \sum_{k=1}^{\infty} 2^{-k} d_H(\mathcal{S}_{k,L}(A), \mathcal{S}_{k,L}(B)). \tag{5}$$

Intuitively, we allow $d_H$ to grow subexponentially in $k$ by the scaling $2^{-k}$. Our definition of profile slightly differs from that of Backhausz and Szegedy (2022), using $L^{\infty}_{\mathrm{reg}(L)}$ instead of their $L^{\infty}_{[-1,1]}$. However, we will justify this deviation in Section 4.4, Theorem 4: by letting $L$ grow slowly in $n$, we recover the original limits in Backhausz and Szegedy (2022).

This *action convergence* turns out to be one of the 'right' notions of convergence that capture both sparse and dense graph limits, as well as some intermediate density graphs:

**Theorem 1** (Theorem 1.1 Backhausz and Szegedy (2022)). *Convergence under $d_M$ is equivalent (results in the same limit) to dense graph convergence when restricted to graphons and equivalent to local-global convergence when restricted to graphings.*

## 3 Graphop neural networks

Graph limits allow us to lift finite graphs onto the richer space of graphops to discuss convergent graph sequences $G_i \to G$. For finite GNNs (Eqn (2)), fixing the graph input $G_i$ and learnable parameter $h$ results in a function $\Phi_F(h, G_i, \cdot)$ that transforms the input graph signal (node features) into an output graph signal. The transferability question asks how similar $\Phi_F(h, G_i, \cdot)$ is to $\Phi_F(h, G_j, \cdot)$ for some $i \neq j$. In our approach using approximation theory, we will compare both functions to the limiting function on $G$. This is done by an appropriate lift of the GNN onto a larger space that we call *graphop neural networks*.

We then introduce a discretization scheme of graphop neural networks to obtain finite GNNs, similar to graphon sampling (Ruiz et al., 2023a) and sampling from topological spaces (Levie et al., 2022). Finally, Lemma 1 asserts that, restricted to self-adjoint $P$-operators, discretizations of graphops are indeed graph shift operators (GSOs).

### 3.1 Convolution and graphop neural networks

Similar to how GSOs in a GNN act on graph signals, graphops act on some $L^2$ signals (called *graphop signals*). The generalization is straightforward: replacing GSOs in the construction of the GNN in Eqn. (2) with graphops results in *graphop convolution* and replacing graph convolution with graphop convolution gives *graphop neural networks*.

Formally, fix a maximum order $K \in \mathbb{N}$. For some measure space $(\Omega, \mathcal{B}, \mu)$, select a graphop $A : L^2(\Omega) \to L^2(\Omega)$ and a graphop signal $X \in L^2(\Omega)$. We define a *graphop convolution* operator as a weighted sum of at most $K - 1$ applications of $A$: $H(h, A)[X] := \sum_{k=0}^{K-1} (h_k A^k)[X]$, where $h \in \mathbb{R}^K$ are (learnable) filter parameters and $A^k$ is the composition of $k$ duplicates of $A$. The square bracket $[v]_i$ indicates the $i$-th entry of a tuple $v$.

For some number of layers $L \in \mathbb{N}, \{n_i\}_{i \in [L]} \in \mathbb{N}, n_0 := 1$, define a *graphop neural network* $\Phi$ with $L$ layers and $n_i$ features in layer $i$ as:

$$\Phi(h, A, X) = X_L(h, A, X), \tag{6}$$

$$X_l(h, A, X) = \left[ \rho\left( \sum_{g=1}^{n_{l-1}} H(h^l_{f,g}, A)[X_{l-1}]_g \right) \right]_{f \in [n_l]}, \qquad l \in [L], \tag{7}$$

$$X_0(h, A, X) = X \tag{8}$$

with filter parameter tuple $h = (h^1, \ldots, h^L)$, $h^l \in (\mathbb{R}^K)^{n_l \times n_{l-1}}$ for any $l \in [L]$, and graphop signal tuple $X_l \in (L^2(\Omega))^{n_l}$ for any $l \in [L] \cup \{0\}$. Eqn (7) and Eqn (2) are almost identical, with the only difference being the input/output space: graphops replacing finite graphs, and graphop signals replacing graph signals.

## 3.2 From graphop neural networks to finite graph neural networks

We are specifically interested in finite GNNs that are discretizations of a graphop (for instance finite grids as discretizations of infinite grids), so as to obtain a quantitative bound that depends on the resolution of discretization. To sample a GNN from a given graphop $A : \mathcal{F} \to \mathcal{F}$, we first sample a GSO and plug it into Eqn (2). Choose a resolution $m \in \mathbb{N}$ and define the GSO $A_m$, for any graph signal $X \in \mathcal{F}_m$ (defined in Section 2) as:

$$A_m X(v) := m \int_{v-\frac{1}{m}}^{v} (A\widetilde{X}) \mathrm{d}\lambda, \qquad v \in [m]/m, \tag{9}$$

$$\Phi_m(h, A, X) := \Phi(h, A_m, X), \tag{10}$$

where graphop signal $\widetilde{X} \in \mathcal{F}$ is an extension of graph signal $X \in \mathcal{F}_m$ defined as

$$\widetilde{X}(u) := X\left(\frac{\lceil um \rceil}{m}\right), \qquad u \in [0,1]. \tag{11}$$

**Shared structural properties of sampled graphs** The discretization scheme introduced for graphop can be intuitively understood as partitioning the vertex set into finitely many sets and merging nodes and their connections in these sets, with an appropriate scaling of the edge weights. Imagine blurring an $n \times n$ matrix into an $n/2 \times n/2$ matrix by average-pooling over each disjoint $2 \times 2$ square. As $n \to \infty$ (and even for uncountable $[0,1]$), this makes rigorous the notion of making a high-resolution graph on a huge number of vertices more 'blurry' by merging nodes in a way that still maintains some smoothness conditions, which is reminiscent of the real-world procedures of training with low-resolution images before fine-tuning with higher-resolution ones, or sampling from low-frequency graph Fourier transform domain.

Note that if $A$ is linear then $A_m$ is necessarily linear, but our definition of graphop does not require linearity. Therefore, $A_m$ is strictly more general than the matrix representation of graph shift operators. We have the following well-definedness result:

**Lemma 1.** *If a graphop $A : \mathcal{F} \to \mathcal{F}$ is self-adjoint, then for each resolution $m \in \mathbb{N}$, the discretization $A_m : \mathcal{F}_m \to \mathcal{F}_m$ defined above is also self-adjoint.*

The proof can be found in Appendix B. Compared to previous works, our discretization scheme in Eqn (9) looks slightly different. In Ruiz et al. (2023a), given a graphon $W : [0,1]^2 \to \mathbb{R}$, the discretization at resolution $n$ was defined by forming the matrix $S \in \mathbb{R}^{n \times n} : S_{i,j} = W(i/n, j/n)$. A related discretization scheme involving picking the interval endpoints at random was also used, but the resulting matrix still takes values at discrete points in $W$. These two sampling schemes rely crucially on their everywhere continuous assumptions for the graphon $W$. Indeed, but for continuity requirements, two functions that differ only at finite discrete points $(i/n, j/n), i, j \in [n]$ are in the same $L^2$ class of functions, but will give rise to completely different samples. Furthermore, not every $L^2$ class of functions has a continuous representative. This means that our discretization scheme is strictly more general than that used by Ruiz et al. (2023a) even when restricted to graphons. This difference comes from the fact that we are discretizing an operator and not the graph itself. For our purpose, taking values at discrete points for some limiting object of sparse graphs will likely not work, since sparsity ensures that most discrete points are trivial.

## 4 Main result: Approximation and transferability

### 4.1 Results for $P$-operators

Our first set of theorems address approximation and transferability of $P$-operators: under certain regularity assumptions to be discussed later, $P$-operators are well approximated by their discretizations:

**Theorem 2** (Approximation theorem)**.** *Let $A : \mathcal{F} \to \mathcal{F}$ be a $P$-operator satisfying Assumption 2 with constant $C_A$; Assumption 3.A or 3.B with resolutions in $\mathcal{N}$. Fix $n \in \mathcal{N}$ and consider $(k, C_v)$-profiles. Let $A_n : \mathcal{F}_n \to \mathcal{F}_n$ be a discretization of $A$ as defined in Eqn (9). Then:*

$$d_M(A, A_n) \leq 2\sqrt{\frac{C_A C_v}{n}} + \frac{C_v + 1}{n}. \tag{12}$$

Compared to theorems in Ruiz et al. (2023a), our explicit dependence on $n$ has an extra $n^{-1/2}$ term that stems from techniques used to bound the Lévy-Prokhorov distance between two entry distributions obtained from functions that differ by at most $O(n^{-1})$ in $L^2$ norm.

As an immediate corollary, invoking the triangle inequality yields a transferability bound.

**Corollary 1** (Transferability). *Let $A : \mathcal{F} \to \mathcal{F}$ be a P-operator satisfying assumptions of Theorem 2 with constant $C_A$ and resolutions $\mathcal{N}$. For any $n, m \in \mathcal{N}$, let $A_n : \mathcal{F}_n \to \mathcal{F}_n$ and $A_m : \mathcal{F}_m \to \mathcal{F}_m$ be discretizations as defined in Eqn (9). Then:*

$$d_M (A_m, A_n) \leq \left( m^{-\frac{1}{2}} + n^{-\frac{1}{2}} \right) 2\sqrt{C_A C_v} + (m^{-1} + n^{-1})(C_v + 1). \tag{13}$$

We emphasize that these theorems work for general nonlinear $P$-operators and not only the linear graphops defined in (Backhausz and Szegedy, 2022).

**Proof sketch**  The full proof of Theorem 2 is in Appendix D. To bound the distance in $d_M$ between two operators, for each sample size $k \in \mathbb{N}$, we give a bound on the Hausdorff metric $d_H$ between the two $(k, C_v)$-profiles. As long as the dependence on $k$ of these bounds is polynomial, the infinite sum in the definition of $d_M$ converges. We do this by picking an arbitrary distribution $\overline{\eta}$ from $\mathcal{S}_{k,C_v}(A)$, which by definition is given by a $k$-tuple $F$ of functions in $L^\infty_{\text{reg}(C_v)}$. Discretize each element of $F$ and consider its entry distribution results in $\overline{\eta}_n \in \mathcal{S}_{k,C_v}(A_n)$. We show that we can give an upper bound of $d_{LP}(\overline{\eta}, \overline{\eta}_n)$ that is independent of the choice of $\overline{\eta}$ and thus same upper bound holds for $\sup_{\eta \in \mathcal{S}_{k,C_v}(A)} \inf_{\eta_n \in \mathcal{S}_{k,C_v}(A_n)} d_{LP}(\eta, \eta_n)$. By also selecting an arbitrary element of $\mathcal{S}_{k,C_v}(A_n)$ and extending it to an element of $\mathcal{S}_{k,C_v}(A)$, we obtain another upper bound for $\sup_{\eta_n \in \mathcal{S}_{k,C_v}(A_n)} \inf_{\eta \in \mathcal{S}_{k,C_v}(A)} d_{LP}(\eta, \eta_n)$ and thus for $d_H$. The different assumptions come in via different techniques used to bound $d_{LP}$ by a high probability bound on the $L^2$ norm of the functions in $F$ and their discretization/extension.

## 4.2  Results for graphop neural networks

Not only are graphops and their discretizations close in $d_M$, but, as we show next, neural networks built from a graphop are also close to those built from graphop discretizations in $d_M$. We iterate that here we are comparing nonlinear operators (graphop neural networks) that are acting on different spaces ($L^2([n]/n)$ for some finite $n$ versus $L^2([0,1])$).

Before stating theoretical guarantees for graphop neural networks, let us introduce some assumptions on the neural network activation function and parameters:

**Assumption 1.** *Let the activation function $\rho : \mathbb{R} \to \mathbb{R}$ in the definition of graphop neural networks be 1-Lipschitz and the convolution parameters $h$ be such that $|h| \leq 1$ element-wise.*

**Theorem 3** (Graphop neural network discretization). *Let $A : \mathcal{F} \to \mathcal{F}$. Assume that $A$ satisfies Assumption 2 with constant $C_A$ and Assumption 3.A or 3.B with resolutions in $\mathcal{N}$. Fix $n \in \mathcal{N}$ and consider $(k, C_v)$-profiles. Under Assumption 1, we have:*

$$d_M(\Phi(h, A, \cdot), \Phi(h, A_n, \cdot)) \leq P_1 \sqrt{\frac{\overline{C}_A C_v}{n}} + \frac{C_v + 1}{n}, \tag{14}$$

*where $\overline{C}_A := (n_{\max} \sum_{i=1}^K C_A^i)^L$, $n_{\max} = \max_{l \in [L]} n_l$, and $P_1$ is a constant depending on $K, L$.*

*Furthermore, we can invoke the triangle inequality to compare outputs of graphop neural networks built from two different discretizations of $A$. For any $m, n \in \mathcal{N}$,*

$$d_M(\Phi(h, A_m, \cdot), \Phi(h, A_n, \cdot)) \leq P_1 \sqrt{\overline{C}_A C_v} \left( m^{-\frac{1}{2}} + n^{-\frac{1}{2}} \right) + (C_v + 1) \left( n^{-1} + m^{-1} \right). \tag{15}$$

Compared to the main theorems of Ruiz et al. (2023a), there are two main differences in our results. First, our rate of $O(n^{-1/2})$ is slower than the rate of $O(n^{-1})$ in Ruiz et al. (2023a) as a function of $n$. Yet, second, their bounds contain a small term that is independent of $n$ and does not go to 0 as $n$ goes to infinity. This small term depends on the variability of small eigenvalues in the spectral decomposition of the convolution operator associated with

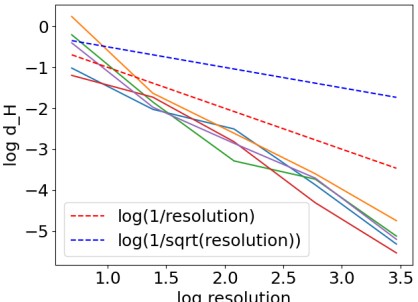

Figure 1: Hausdorff metric between samples from 1-profiles of 2-hidden-layer GNN on finite polymer graphs vs on large polymer graphs (see Appendix A for polymer graphs). The GNN uses GSO $A_n^2 + A_n$ where $A_n$ is the normalized adjacency matrix on $n$ nodes and ReLU nonlinearties at each layer. Different solid lines are different random draws of functions that make up the estimated 1-profile. See Appendix A for details.

a graphon. The bound in Theorem 3, in contrast, goes to zero. We tested this bound in Figure 1 for GNNs on polymer graphs, which suggest that $O(n^{-1})$ rate may be possible.

The proof for this theorem is in Appendix D.3 for a more general Theorem 6. Note that it does not suffice to simply use the fact that the assumptions play well with composition with Lipschitz function $\rho$, which would result in a bound involving $\Phi(h, A, \cdot)$ and its discretization $(\Phi(h, A, \cdot))_n$ as a nonlinear operator, as opposed to a bound between $\Phi(h, A, \cdot)$ and $\Phi(h, A_n, \cdot)$. Our proof shares the same structure as that of Theorem 2 while making sure that the mismatch from discretizing/extending operators does not blow up with composition.

### 4.3 Assumptions

We discuss the main assumptions of our $P$-operators.

**Assumption 2** (Lipschitz mapping). *An operator $A : \mathcal{F} \to \mathcal{F}$ is $C_A$-Lipschitz if $\|Af - Ag\|_2 \leq C_A \|f - g\|_2$ for any $f, g \in \mathcal{F}$.*

We have already had a finite bound on the operator norm in the definition of $P$-operators. For linear operators, Assumption 2 is equivalent to a bounded operator norm and is thus automatically satisfied by linear $P$-operators.

The next few assumptions are alternatives; only one needs be satisfied by our $P$-operators. Intuitively, they ensure that the images of our operator are not too discontinuous:

**Assumption 3.A** (Maps constant pieces to constant pieces). *We say that an operator $A : \mathcal{F} \to \mathcal{F}$ maps constant pieces to constant pieces at resolutions in $\mathcal{N} \subset \mathbb{N}$ if for any $n \in \mathcal{N}$, and for any $f \in \mathcal{F}_{[-1,1]}$ that is a.e. constant on each interval $(u - 1/n, u]$ for $u \in [n]/n$, $Af$ is also constant on $(u - 1/n, u]$ for each $u$.*

**Assumption 3.B** (Maps Lipschitz functions to Lipschitz functions). *We say that an operator $A : \mathcal{F} \to \mathcal{F}$ maps Lipschitz functions to Lipschitz functions at resolutions in $\mathcal{N} \subset \mathbb{N}$ if for any $n \in \mathcal{N}$, and for any $f \in \mathcal{F}_{reg(C_v)}$, $Af$ is $C_v$-Lipschitz.*

This is the most restrictive assumption. However, the next lemma (proof in Appendix D) describes some dense, sparse and intermediate graphs that satisfy these assumptions.

**Lemma 2** (Well-behaved operators). *The following examples satisy our assumptions:*

1. Bounded-degree graphings: *Let $G$ be a graphing corresponding $k$-D grids or polymer graphs (not necessarily regular graphs) with fixed monomers (Appendix A). For each $N \in \mathbb{N}$, there exists a locally equivalent graphing $G'_N$ such that its adjacency operator satisfies Assumption 3.A with some resolution set (see Lemma 5 and Lemma 7).*

2. Lipschitz graphons: *Let $W$ be a $C_v$-Lipschitz graphon on $\mathcal{F}_{reg(C_v)}$. Then the Hilbert-Schmidt operator $f \mapsto \int_0^1 W(\cdot, y) g(y) \mathrm{d}y$ satisfies Assumption 3.B with resolution in $\mathbb{N}$.*

3. General graphs: *Let $G$ be a (potentially infinite) graph with a vertex coloring of $N$ colors such that if two vertices have the same color, then the multisets of their neighbors' colors are the same. Then $G$'s adjacency operator satisfies Assumption 3.A with resolution $N$. An $N$-d hypercube (more generally, Hamming graphs) which is neither bounded-degree nor dense, satisfies the above condition with resolutions in $\{2^n\}_{n \in [N]}$.*

All our results also hold with a less restrictive assumption that allows for a failure of Assumption 3.A and 3.B in a small set (see Assumption 4.A and 4.B in the Appendix). The most general results are proven in Appendix D and hold in even slightly more relaxed conditions which require the operators to map constant pieces to *Lipschitz* pieces (Assumption 5.A, 5.B in Appendix D).

## 4.4 Deviations and Justifications

All our theorems hold in slightly modified settings than those by Backhausz and Szegedy (2022). Namely, we allowed for nonlinear $P$-operators, assumed that they have finite $\|\cdot\|_{2\to 2}$ norm, and used $(k, L)$-profiles where we focus on Lipschitz functions (while Backhausz and Szegedy (2022) consider all measurable functions in their profiles). Therefore, we need to ensure that our changes still give us a useful mode of convergence that generalizes dense and sparse graph convergences.

First, without the linearity assumption, the convergence proof by Backhausz and Szegedy (2022) does not hold: we do not know if all limits of nonlinear graphops are still graphops. However, our approximation results (Theorem 2) show special convergent sequences of nonlinear operators, which go beyond the settings in (Backhausz and Szegedy, 2022). Studying special nonlinear operator sequences is interesting since graphop NNs themselves are nonlinear operators. We also assert that our restriction to operators acting on $L^2$ spaces does not affect convergence guarantees (Theorem 2.14 in (Backhausz and Szegedy, 2022)).

Next, we show that restriction to Lipschitz profiles, which is necessary for our proof technique, does not affect the original convergence either, if we allow our Lipschitz constant to grow:

**Theorem 4** (Growing profiles). *Let $L : \mathbb{N} \to \mathbb{R}$ be a strictly increasing sequence such that $L(n) \xrightarrow{n\to\infty} \infty$. Consider a sequence of $P$-operators $(A_n : \mathcal{F}_n \to \mathcal{F}_n)_{n\in\mathbb{N}}$ that is Cauchy in the sense that $d'_M(A_n, A_m) := \sum_{k=1}^{\infty} 2^{-k} d_H(\mathcal{S}_{k,L(n)}(A_n), \mathcal{S}_{k,L(m)}(A_m)) \to 0$ as $m, n \to \infty$. If $A_n \to A$ under action convergence (Backhausz and Szegedy, 2022), then $(A_n)_{n\in\mathbb{N}}$ converges to the same limit under $d'_M$.*

This theorem allows us to replace the $C_v$ constant in our bound with a slowly growing function in $n$ and get back 'action convergence' as described in Backhausz and Szegedy (2022) so that we can inherit its useful properties while still be able to draw on Lipschitz assumptions in the profiles, without any realistic slowdown in the bound.

**Proof sketch** For some $k \in \mathbb{N}$, by the completeness of the Hausdorff metric over the closed subsets of the space of probability measures supported on $\mathbb{R}^{2k}$, the statement is equivalent to showing $d_H(\mathcal{S}_k(A), \mathcal{S}_{k,L(n)}(A_n)) \to 0$ as $n \to \infty$. The proof uses a Lipschitz mollification argument to smooth out arbitrary measurable functions $f_1, \ldots, f_k$ that witness a measure in the $k$-profile of $A$. By selecting a Lipschitz mollifier $\phi$, we ensure that convolving $f_j$ with $\phi_\epsilon : x \mapsto \epsilon^{-1}\phi(x\epsilon^{-1})$ results in a Lipschitz function that converges to $f$ in $L^2$ as $\epsilon$ goes to 0.

## 5 Discussion and Future directions

In this paper, we study size transferability of finite GNNs on graphs that are discretizations of graphop, a recent notion of graph limit introduced by Backhausz and Szegedy (2022). We achieve this by viewing GNNs as operators that transform one graph signal into another. Under regularity assumptions, we proved that two GNNs, using two different-resolution GSOs discretized from the same graphop, are close in an operator metric built from weak convergence of measures.

For future direction, a principled study of spectral properties of graphops and graphop neural networks would open doors for techniques from Fourier analysis as used in (Ruiz et al., 2023a; Levie et al., 2022). This leads to distinct challenges, e.g., the spectral gap is not continuous with respect to local-global limits and thus action convergence, but many more properties of spectral measures of bounded-degree graphs are recently studied (Virag, 2018).

## Acknowledgments and Disclosure of Funding

This work was supported by Office of Naval Research grant N00014-20-1-2023 (MURI ML-SCOPE), NSF award CCF-2112665 (TILOS AI Institute), NSF award 2134108.

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

# A  Additional details

Some highly repetitive parts in the proofs are presented informally (e.g. "use the same techniques as...") to highlight the main ideas in the NeurIPS 2023 version of the paper. For the complete formal proof, readers are advised to study a later version on arXiv.

## A.1  Notations

We will use the following extra notations in the proof:

1. For some $P$-operator $A \in L^2(\Omega) \to L^2(\Omega)$, $k \in \mathbb{N}$ and $\{f_1, \ldots, f_k\} =: F \subset L^2(\Omega)$, let $F_A$ be the ordered $2k$-tuple $(f_1, \ldots, f_k, Af_1, \ldots, Af_k)$ and denote

$$F_A(x) := (f_1(x), \ldots, f_k(x), Af_1(x), \ldots, Af_k(x)) \in \mathbb{R}^{2k.}$$

2. For some $P$-operator $A \in L^2(\Omega) \to L^2(\Omega)$, $k \in \mathbb{N}$ and $\{f_1, \ldots, f_k\} =: F \subset L^2(\Omega)$, let $\mathcal{D}_A(F)$ be the entry distribution $\mathcal{D}(f_1, \ldots, f_k, Af_1, \ldots, Af_k)$ .

## A.2  Definitions

**Cut norm and cut metric**   Here we define concretely the cut metric over the space of graphons. Recall that graphons are $L^1([0,1], \mathcal{F}, \lambda)$ Lebesgue-integrable functions. The space of graphons is equipped with a norm known as the *cut norm*:

$$\|W\|_\square := \sup_{S,T \in \mathcal{F}} \left| \int_{S \times T} W(x,y) \mathrm{d}\lambda(x) \mathrm{d}\lambda(y) \right|, \tag{16}$$

and a metric known as the *cut metric*:

$$d_\square(W_1, W_2) := \inf_\phi \|W_1 - W_2 \circ \phi\|_\square, \tag{17}$$

where $\phi$ is taken over all measure-preserving bijections from $[0,1]$ to $[0,1]$. Intuitively, taking the inf over all measure-preserving bijections allows the cut metric to identify graphons that are just a rearrangement away from another. This generalizes symmetries in graphs, where permuting the vertices (and the corresponding edges) do not change the graph itself. Lovász (2012) shows that graphon convergence under the cut metric is well-behaved: every graphon is a limit of a convergent sequence of graphons; and every Cauchy sequences converges to a graphon. This mode of convergence is known as dense graph convergence.

**Lévy-Prokhorov metric**   The definition of Lévy-Prokhorov metric on $\mathcal{P}(\mathbb{R}^{2k})$ is:

$$d_{LP}(\eta_1, \eta_2) := \inf\{\epsilon > 0 : \eta_1(U) \le \eta_2(U^\epsilon) + \epsilon \wedge \eta_2(U) \le \eta_1(U^\epsilon) + \epsilon, \forall U \in \mathcal{B}_{2k}\},$$

where $\mathcal{B}_{2k}$ is the Borel $\sigma$-algebra generated from open subsets of $\mathbb{R}^{2k}$ and

$$U^\epsilon := \{y : \exists x \in \mathbb{R}^{2k} \|x - y\|_2 < \epsilon\}. \tag{18}$$

**Specific graphs**   We will also define different graphs that we mentioned in the main paper. Unless otherwise stated, we consider undirected graphs. A *path* on $n$-vertices for some $n \ge 2$ is the graph $G = ([n], \{(i, i+1) : i \in [n-1]\})$. Its high dimensional generalization are *k-D grids* on $n^k$ vertices are the graphs $G_k = ([n]^k, \{(u,v) : \|u-v\|_1 = 1\})$.

A *Hamming graph* $H(d,q)$ with parameters $d, q \ge 1 \in \mathbb{N}$ is the graph with vertex set $[q]^d$ and edge set contains all pairs $u, v$ that differs at exactly 1 coordinate (Hamming distance 1). For instance, $H(1,q)$ is the complete graph on $q$ vertices and $H(2,q)$ is the hypercube on $2^q$ vertices.

An *polymer graph* $P_k$ with $k \ge 2$ and finite $n$-vertex monomer $M = (V_M, E_M)$ is the graph on $kn$ vertices that contains $k$ copies $M_1, \ldots, M_k$ of $M$. Since $M$ is finite, we can choose a start $s_1$ and end $e_1$ vertices in $V_{M_1}$. $M_i$'s being copies of $M$ also means that there is an isomorphism $\phi_i$ from $V_{M_{i-1}}$ to $V_{M_i}$ for each $i \in [k-1]$. Let $s_i$ and $e_i$ be inductively defined as $\phi_i(s_{i-1})$ and $\phi_i(e_{i-1})$ for $i$ from 2 to $k$. Beside edges in the copies of $M$, we also add all edges of the form $(e_{i-1}, s_i)$ for each $i$ from 2 to $k$. See Figure 2 for an illustration.

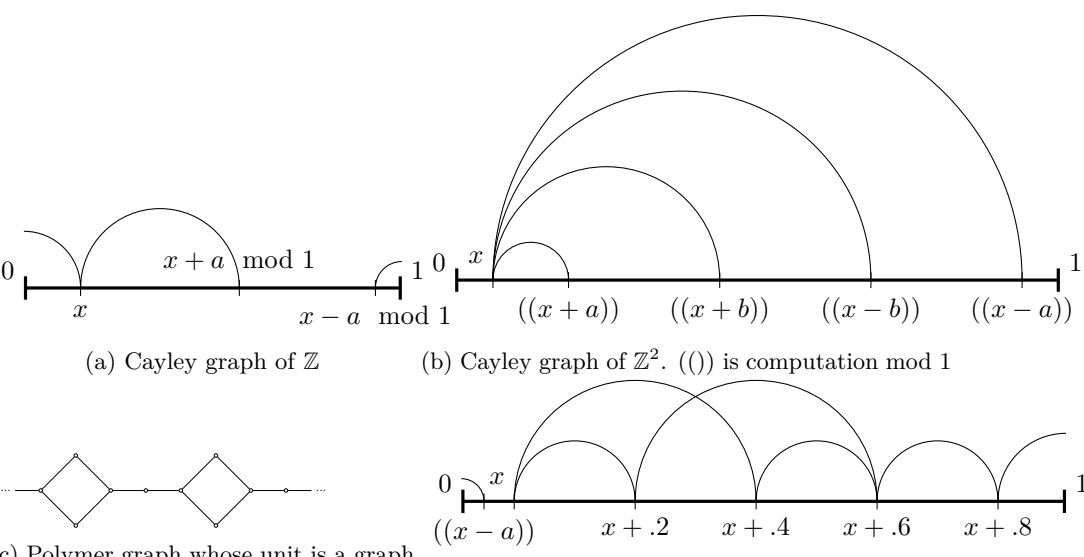

(a) Cayley graph of $\mathbb{Z}$        (b) Cayley graph of $\mathbb{Z}^2$. (()) is computation mod 1

(c) Polymer graph whose unit is a graph with 5 vertices. The pattern extends ad infinitum on both sides.   (d) Graphing of (c). (()) is computation mod 0.2. Edges drawn are from a single unit in the polymer.

Figure 2: Examples of limit objects. The vertex set is the interval $[0, 1]$. Example edges are the arcs connecting points on the intervals. $a$ and $b$ are distinct irrational numbers. In each graph, edges that miss an endpoint are identified as a single edge connecting the two existing endpoints.

## A.3 Illustrations

Figure 2 shows examples of graphings for a Cayley graph of $\mathbb{Z}, \mathbb{Z}^2$ and a polynomer graph.

## A.4 Experimental details

To empirically study the rate of convergence of the $d_M$ metric, we study an approximation of Hausdorff metric between 1-profiles of GNNs build on small finite polymers to that built on a large one. The result is in Figure 1 and we now detail the set up.

**Architecture**    The GNN has 2 hidden layers, uses ReLU activation at each layer, including the output layer. We fix a particular polynomial GSO $A^2 + A$ where $A$ is the normalized (by edge vertex degree) adjacency matrix of the input graph of the GNN. Monomer for each polymer has size 5 and is the same monomer seen in Figure 2. The small polymers consist of $2, 4, 8, 16, 32$ monomers. While the large monomers that take the place of the limit object has 128 monomers.

1**-profile estimation**    Recall that the 1-profile of an operator $P$ consists of entry distributions of $(f, Pf)$ where $f$ runs through all measurable functions. It is therefore, not tractable to construct the 1-profile exactly. Instead, we draw random $F = \{f_1, f_2, \ldots, f_{10}\}$'s to construct a set of 10 probability distributions. Repetitions of this random drawing make up the different solid lines in Figure 1. We construct each $f_i : [0, 1] \to [-1, 1]$ as piecewise linear function with pieces $((u - 1)/1000, u/1000]$ for $u$ from 1 to 1000. We set $f_i(0) = 0$ for each $i$ and recursively set $f_i(u/1000) = f_i((u - 1)/1000) + q/100$ where $q$ is a random draw from $\{-1, 1\}$ uniformly. Finally, we linearly interpolate between two consecutive endpoints. Numpy random seed is set at 1234567.

After getting this set $F$, for each operator $P$ that are either GNN built on small polymers or GNN built on the large one, we compute the set $\{\mathcal{D}(f_i, Pf_i) : f_i \in F\}$ to get the corresponding estimation of the 1-profile.

**Hausdorff metric computation** Given a pair of 1-profile estimations to compute $d_H$ over, we use optimal transport code from (Flamary et al., 2021) to compute the earth mover distance between each pairs of distributions. This gives an approximation of $d_{LP}$ between elements of the two profiles. Using the definition of Hausdorff distance, we obtain the result in Figure 1.

### A.5 Milder assumptions

As mentioned in the main text, we will work with the following slightly less restrictive set of Assumptions.

**Assumption 4.A** (Maps constant pieces to constant pieces with high probability). *We say that an operator $A : \mathcal{F} \to \mathcal{F}$ maps constant pieces to constant pieces whp at resolutions in $\mathcal{N} \subset \mathbb{N}$ if there exists a set $E \subset [0,1]$ with Lebesgue measure $\lambda(E) < \inf_{n \in \mathcal{N}} 1/n$, such that for any $n \in \mathcal{N}$, and for any $f \in \mathcal{F}_{[-1,1]}$ that is a.e. constant on each interval $(u - 1/n, u]$ for $u \in [n]/n$, $Af$ is constant on $(u - 1/n, u] \backslash E$ and $\|Af\mathbb{1}_E\|_1 < \inf_{n \in \mathcal{N}} \frac{1}{n}$.*

**Assumption 4.B** (Maps Lipschitz functions to Lipschitz functions with high probability). *We say that an operator $A : \mathcal{F} \to \mathcal{F}$ maps Lipschitz functions to Lipschitz functions whp at resolutions in $\mathcal{N} \subset \mathbb{N}$ if there exists a set $E \subset [0,1]$ with $\lambda(E) < \inf_{n \in \mathcal{N}} 1/n$, such that for any $n \in \mathcal{N}$, and for any $f \in \mathcal{F}_{[-1,1]}$, $Af$ is $C_v$ Lipschitz on $[0,1] \backslash E$ and $\|Af\mathbb{1}_E\|_1 < \inf_{n \in \mathcal{N}} \frac{1}{n}$.*

**Assumption 5.A** (Maps constant pieces to Lipschitz pieces). *We say that an operator $A : \mathcal{F} \to \mathcal{F}$ maps constant pieces to Lipschitz pieces at resolutions in $\mathcal{N} \subset \mathbb{N}$ and constant $C$ if for any $n \in \mathcal{N}$, and for any $f \in \mathcal{F}_{[-1,1]}$ that is a.e. constant on each interval $(u - 1/n, u]$ for $u \in [n]/n$, we have that $Af$ is $C$-Lipschitz on each $(u - 1/n, u]$, for all $u \in [n]/n$.*

**Assumption 5.B** (Maps constant pieces to Lipschitz pieces with high probability). *We say that an operator $A : \mathcal{F} \to \mathcal{F}$ maps constant pieces to Lipschitz pieces whp at resolutions in $\mathcal{N} \subset \mathbb{N}$ and constant $C$ if for any $n \in \mathcal{N}$, there exists a set $E \subset [0,1]$ with $\lambda(E) < \frac{1}{n}$ such that for any $f \in \mathcal{F}_{[-1,1]}$ that is a.e. constant on each interval $(u - 1/n, u]$ for $u \in [n]/n$, it holds that $Af$ is $C$-Lipschitz on each $(u - 1/n, u] \backslash E$, for all $u \in [n]/n$ and $\|Af\mathbb{1}_E\|_1 < \inf_{n \in \mathcal{N}} \frac{1}{n}$.*

## B  Omitted proofs from Section 3

*Proof of Lemma 1.* Fix $m \in \mathbb{N}$ and $f, g \in \mathcal{F}_m$. Since $P$-operators are bounded, to show that they are self-adjoint, it suffices to show that $\langle A_m f, g \rangle = \langle f, A_m g \rangle$ where $\langle \cdot, \cdot \rangle$ is the usual inner product in the Hilbert space $\mathcal{F}_m$. We have:

$$\langle A_m f, g \rangle = \sum_{u \in 1/m[m]} (A_m f)(u) g(u) \tag{19}$$

$$= \sum_{u \in 1/m[m]} \int_{u - \frac{1}{m}}^{u} A f' \mathrm{d}\lambda \cdot g(u) \tag{20}$$

$$= \sum_{u \in 1/m[m]} \int_{u - \frac{1}{m}}^{u} (A f') g' \mathrm{d}\lambda = \int_{0}^{1} (A f') g' \mathrm{d}\lambda \tag{21}$$

$$= \int_{0}^{1} f'(A g') \mathrm{d}\lambda \tag{22}$$

$$= \sum_{u \in 1/m[m]} \int_{u - \frac{1}{m}}^{u} f'(A g') \mathrm{d}\lambda = \langle f, A g \rangle, \tag{23}$$

where the first line is the definition of the inner product in $\mathcal{F}_m$, the second line is the definition of the discretization $A_m$ (recall that for $f \in \mathcal{F}_m$, $f' \in \mathcal{F}$ is the extension of $f$ defined as $f'(x) = f(\lceil xm \rceil /m)$), the third line is because $g'$ is constant on fixed $[u - 1/m, u]$ intervals for each $u$ and the fourth line is because $A$ is self-adjoint.  $\square$

# C Theory of graphings

In this subsection, we highlight definitions and key characteristics of graphings so that the paper is self-contained. A much more in-depth discussion can be found in Lovász (2012).

**Definition 1** (Borel graphs). *Let $\Omega$ be a topological space and $(\Omega, \mathcal{F})$ be the corresponding Borel space. A* Borel graph *is a graph $(\Omega, E)$ such that $E \in \mathcal{F} \times \mathcal{F}$.*

The following proposition asserts that bounded-degree graphs without automorphisms are always Borel:

**Lemma 3** (Proposition 18.6 from Lovász (2012)). *If $G$ is a bounded-degree graph without automorphisms then there exists a topology $\tau$ on $V(G)$ (called the* local topology*) such that $G$ is Borel with respect to the Borel space built from $\tau$.*

When the graph does have automorphisms, one can break the symmetries by coloring the nodes with some set of colors (the fact that the graph has bounded degree means that one only needs finitely many colors).

We next introduce the main object of interest:

**Definition 2** (Graphings). *A* graphing *is a quadruple $G = (\Omega, \mathcal{F}, \lambda, E)$ such that $\mathcal{F}$ is a Borel $\sigma$-algebra that makes $(\Omega, E)$ a Borel graph, and $\lambda$ is a probability measure on $(\Omega, \mathcal{F})$ satisfying: for any $A, B \in \mathcal{F}$,*

$$\int_A deg_B(x)\mathrm{d}\lambda(x) = \int_B deg_A(x)\mathrm{d}\lambda(x), \tag{24}$$

*where $deg_A(x)$ counts the number of neighbors in $A$ of $x$.*

As an example, we now describe paths and cycles in terms of graphings: let $V$ be $[0, 1]$, $\mathcal{F}$ the Borel $\sigma$-algebra generated by open intervals with rational endpoints and for each $x \in [0, 1]$, put $(x, x \pm a)$ in $E$ if $x \pm a \in [0, 1]$ for some real number $a < 1$. For $a < \frac{1}{2}$, each connected component of the graphing is a finite path. If we consider the edge set $(x, x \pm a \mod 1)$ for rational $a$, then it is not hard to see that connected components of the graphing are finite cycles. If $a$ is irrational, then the resulting graphing is a two-way infinite path, i.e., a path with no "beginning" and no "end". A formal argument, with an appropriate metric on the space of graphings, can be made to show that the limit of cycles and paths coincides to be the two-way infinite path.

Graphings are not unique in representing certain graphs. There are weak equivalences between pairs of graphings that are formalized through the notion of local isomorphisms.

**Definition 3** (Local isomorphisms of graphings). *Let $G_1$, $G_2$ be two graphings. A measure-preserving map $\varphi : V(G_1) \to V(G_2)$ is a* local isomorphism *if its restriction to almost every connected component of $G_1$ (outside of a set of connected component of measure 0) is a graph isomorphism with one of the connect components of $G_2$. More formally,*

$$\Pr_{\lambda(G_1)}((G_1)_x \equiv (G_2)_{\varphi(x)}) = 1, \tag{25}$$

*where $\equiv$ is rooted graph isomorphism and $(G_1)_x$ is the connected component of $G_1$ rooted at $x$.*

Note that local isomorphism of graphings are not symmetric and the map $\varphi$ needs not be invertible. A stronger notion of equivalence, which is symmetric and transitive is local equivalence:

**Definition 4** (Local equivalence (informal)). *$G_1$ and $G_2$ are* locally equivalent *if they have the same subgraph densities $t^*(F, G_1) = t^*(F, G_2)$ for every connected simple graph $F$.*

The above definition is informal since we have not defined subgraph densities (which is done via the Benjamini-Schramm interpretation of graphings). We state Definition 4 for readers who are familiar with dense graph convergence of graphons since this is how such convergences are defined. In fact, we can conveniently bypass formally defining local equivalence by the following characterization:

**Lemma 4** (Bi-local isomorphism (Theorem 18.59 in Lovász (2012))). *Two graphings are locally equivalent iff there is a third graphing with a local isomorphism to each of them - a property called* bi-local isomorphism.

### C.1 Degeneracy of eigenvectors

We give an heuristical argument why directed graphing limits can fail to have eigenvalues and eigenvectors/eigenfunctions. Consider the sequence of one sided directed path: $G_n = (V = [n], E = \{(i, i+1) : i \in [n-1]\})$, for $n \geq 1$. Note that this may not be a Cayley graph in certain authors' definition, which requires Caley graphs to be undirected. However, it is a realistic example in machine learning since convolutional layers in CNNs are exactly the adjacency operators of these directed graphs. The limiting adjacency operator for this sequence, for all intents and purposes, is intuitively the shift operator (say, on $\ell^2$) $A : (x_1, x_2, x_3, \ldots) \mapsto (0, x_1, x_2, \ldots)$. An easy calculation then shows that this shift operator does not have eigenvalues.

For a more formal argument and construction on failure cases of existence of eigenvectors in undirected graphings (which is what we defined in this section), refer to Remark 1.6 of Backhausz and Szegedy (2022).

## D  Omitted proofs from Section 4

### D.1  Proof of Lemma 2

We prove each bullet point in Lemma 2 separately, in the next three lemmas.

In the following lemma, we show that graphings corresponding to infinite paths and high dimensional grids satisfy the assumptions in the main results. Similar to how isomorphic graphs represent the same graph, we only need to specify a locally isomorphic graphing that represent the equivalence class of graphings containing the infinite path and high dimensional grids.

**Lemma 5** (Well-behaved GSOs - Graphings). *Let $G$ be a graphing corresponding to the Cayley graph of $\mathbb{Z}$ (two-way infinite paths) or high-dimensional generalizations (infinite 2D and 3D grids). For a graphing $H$, let $A(H)$ be its adjacency operator. If $H$ is regular, let $deg(H)$ be the degree of any of its vertices. For each $N \in \mathbb{N}$, there exists locally equivalent graphings:*

1. *$G'_N$ such that $A(G'_N)$ satisfies Assumption 3.A with resolution set $d(N) = \{x \in \mathbb{N} : x\alpha = N \text{ for some } \alpha \in \mathbb{N}\}$.*

2. *$G''_N$ such that $A(G''_N)$ satisfies Assumption 4.A with resolution set $[N]$.*

3. *$G'''_N$ such that $A(G'''_N)/deg(G)$ satisfies Assumption 4.B with resolution set $[N]$.*

*Proof of Lemma 5.* We will first show the results for two-way infinite paths. Higher dimensional versions follow almost verbatim. Fix $a \in \mathbb{R}\backslash\mathbb{Q}$ irrational. Recall from Lovász (2012) that the graphing $G = ([0,1], \mathcal{F}, \lambda, E)$ where $\mathcal{F}$ is the Borel $\sigma$-algebra generated by open intervals with rational endpoints, $\lambda$ is some probability measure on $([0,1], \mathcal{F})$ and $E \in \mathcal{F} \times \mathcal{F}$ is defined as:

$$E := \{(x, x \pm a \mod 1) \mid x \in [0,1]\}. \tag{26}$$

That $G$ is a graphing and each of its connected components is a copy of the Cayley graph of $\mathbb{Z}$ generated by $\{-1, 1\}$ is asserted in Lovász (2012).

1. Fix $N \in \mathbb{N}$, the goal is to define a graphing $G'_N$ that is locally equivalent to $G$ such that $A(G'_N)$ satisfies Assumption 3.A with resolution set $d(N)$ - the divisor set of $N$.

   **Defining $G'_N$.**  Let $G'_N := ([0,1], \mathcal{F}, \lambda, E'_N)$ where,

   $$E'_N := \left\{ \left( x_j, j - \frac{1}{N} + \left( x_j \pm \frac{a}{N} \mod \frac{1}{N} \right) \right) : j \in [N]/N, x_j \in \left[ j - \frac{1}{N}, j \right] \right\}. \tag{27}$$

   Intuitively, $G'_N$ consists of $N$ disjoint copies of $G$ shrunk to the space $[0, 1/N)$. Since $E \subset [0,1] \times [0,1]$, $G'_N$ is a graphing in the same Borel space as $G$.

**$G'_N$ is locally equivalent to $G$.** From Lemma 4 it suffices to display a local isomorphism from a third graphing to each of them. Let $G'_{2N}$, defined similarly as $G'_N$, be the third graphing. We claim that $\varphi_1 : x \mapsto 2x \mod 1$ is a local isomorphism from $G'_{2N}$ to $G'_N$ and $\varphi_2 : x \mapsto 2Nx \mod 1$ is a local isomorphism from $G'_{2N}$ to $G$. Intuitively, $G'_{2N}$ contains $2N$ copies of $G$ while $G'_N$ contains $N$ copies of $G$. Thus, our local isomorphisms only need to make sure that a connected component in one copy of $G$ in $G'_N$ is mapped bijectively to a connected component in another copy of $G$ in $G'_{2N}$. We give a rigorous argument below.

For $x$ picked randomly according to $\lambda$, let $j \in [2N]/(2N)$ be such that $x \in [j - 1/(2N), j)$. By definition of $G'_{2N}$, the connected component $(G'_{2N})_x$ consists of vertices of the form

$$v(k) = j - \frac{1}{2N} + (x \pm ak/(2N) \mod 1/(2N)),$$

for some $k \in \mathbb{Z}$ and there is an edge from $v(k)$ to $v(k \pm 1)$. Consider $\varphi_1(x) = 2x \mod 1$ that sends $v(k)$ to

$$\varphi_1(v(k)) := \left( 2j - \frac{1}{N} + \left( 2x \pm \frac{ak}{N} \mod \frac{1}{N} \right) \right) \mod 1.$$

Now consider the connected component of $\varphi_1(x) = 2x \mod 1$ in $G'_N$. Since $x \in [j - \frac{1}{2N}, j), \varphi_1(x) \in [j' - 1/N, j')$ where $[N]/N \ni j' := 2j \mod \frac{1}{N}$. Thus, the connected component of $(G'_N)_{\varphi_1(x)}$ consists of vertices of the form:

$$v'(k) := j' - \frac{1}{N} + (\varphi_1(x) \pm ak/N \mod 1/N) \tag{28}$$

$$= \left( 2j \mod \frac{1}{N} \right) - \frac{1}{N} + \left( (2x \mod 1) \pm \frac{ak}{N} \mod \frac{1}{N} \right). \tag{29}$$

With some modulo arithmetic manipulation, it is not hard to see that $\varphi_1(v(k)) = v'(k)$ for all $k \in \mathbb{Z}$. By definition of $G'_N$, there is an edge $(v'(k), v'(k \pm 1))$. Therefore, there is an edge $(\varphi_1(v(k)), \varphi_1(v(k \pm 1)))$. If $\lambda$ is the uniform measure then we can ignore vertices at the endpoints of our intervals ( points in $[N]/N$ ) and conclude that $(G'_{2N})_x \equiv (G'_N)_{\varphi_1(x)}$ with probability 1 when $x \sim \lambda$.

Now consider $\varphi_2(x) = 2Nx \mod 1$ that sends $v(k)$ to:

$$\varphi_2(v(k)) := (2Nj - 1 + (2Nx \pm ak \mod 1)) \mod 1. \tag{30}$$

The connected component of $\phi_2(x)$ in $G$ is in $[0, 1]$ and consists of vertices of the form:

$$v''(k) := (2Nj - 1 \mod 1) + (2Nx \pm ak \mod 1). \tag{31}$$

Thus $v''(k) = \varphi_2(v(k))$ for each $k \in \mathbb{Z}$ and the definition of $G$ implies that $(G'_{2N})_x \equiv G_{\varphi_2(x)}$ since both are isomorphic to the two-way infinite path. If $\lambda$ is the uniform measure then the above hold for a.e. $x \in [0, 1]$.

**$G'_N$ satisfies Assumption 3.A with resolution set $d(N)$.** Let $D$ be a divisor of $N$, then $D\alpha = N$ for some $\alpha \in \mathbb{N}$. The intuition is rather straightforward, since $D$ is a divisor of $N$, the partition into $N$ equal intervals of $[0, 1]$ is simply a finer partition into $D$ equal intervals. By construction, a vertex $x$ of $G'_N$ only has neighbors in the $1/N$ interval containing it. Therefore, if $f$ is constant on each of the $D$ pieces, it is also constant on each of the $N$ pieces and $f(y) = f(x) = f(x') = f(y')$ for any $(x, y), (x', y')$ neighbors such that $x$ and $x'$ come from the same $1/D$ pieces.

Here is a more formal argument. Fix $f \in \mathcal{F}_{[-1,1]}$ that is a.e. constant on each interval $(u - 1/d, u]$ for $u \in [D]/D$. We need to show that $A(G'_N)f$ is a.e. constant on the same pieces where $A(H)$ is the adjacency operator of a regular graphing $H$. Fix $d \in [D]$, fix $x, x' \in [d - 1/D, d)$. Let $j$ be such that $x \in [j - 1/N, j)$ and $j'$ be

such that $x' \in [j' - 1/N, j')$. Since $D\alpha = N$, we have $[j - 1/N, j), [j' - 1/N, j') \subseteq [d - 1/D, d)$. Furthermore, by definition of $G'_N$, all neighbors of $x$ and $x'$ are in $[j - 1/N, j)$ and $[j' - 1/N, j')$ respectively, and thus in $[d - 1/D, d)$. Therefore, $A(G'_N)f(x) = 2f(x) = 2f(x') = A(G'_N)f(x')$ for all $x, x' \in [d - 1/D, d)$, which means that $A(G'_N)f$ is constant on each of the pieces $[d - 1/D, d)$ for each $d \in [D]$.

2. Fix $N \in \mathcal{N}$, the goal is to define a graphing $G''_N$ that is locally equivalent to $G$ such that $A(G''_N)$ satisfies Assumption 4.A with resolution set $[N]$.

   Since $\mathbb{Q}$ is dense in $\mathbb{R}$, there exists a number $\delta(N) < \frac{1}{4N^2}$ such that $\frac{1}{4N^2} + \delta(N)$ is an irrational number. Let $G''_N = ([0, 1], \mathcal{F}, \lambda, E''_N)$ where,

   $$E''_N := \left\{ \left( x, x \pm \left( \frac{1}{4N^2} + \delta(N) \right) \mod 1 \right) : x \in [0, 1] \right\}. \tag{32}$$

   That $G''_N$ is locally equivalence to $G$ is easily seen via the ambiguity of selecting $a$ when defining $G$. Now we show that $A(G''_N)$ satisfies Assumption 4.A with resolution set $[N]$. Pick $M \leq N$ and $f \in \mathcal{F}_{[-1,1]}$ that is a.e. constant on each pieces $(m - 1/M, m]$ for each $m \in [M]/M$. Let

   $$E = \bigcup_{m \in [M]/M} \left( m - \frac{1}{4N^2} - \delta(N), m + \frac{1}{4N^2} + \delta(N) \right]$$

   then

   $$\lambda(E) = \sum_{m'=1}^{M} 2(1/(4N^2) + \delta(N)) \leq \frac{M}{N^2} < \frac{1}{N}.$$

   Pick $x, x' \in (m - 1/M, m] \setminus E$ then $x, x' \in (m - 1/M + \epsilon, m - \epsilon]$ where $\epsilon = 1/(4N^2) + \delta(N)$. Thus we have both $x \pm \epsilon$ and $x' \pm \epsilon$ are in $(m - 1/M, m]$. By definition of $G''_N$, all neighbors of $x$ and $x'$ are in the same $1/M$ piece as $x$ and $x'$. Therefore, if $f \in \mathcal{F}_{[-1,1]}$ are constant on these pieces, so is $Af$.

3. Fix $N \in \mathcal{N}$, the goal is define a graphing $G'''_N$ that is locally equivalent to $G$ such that $A(G'''_N)$ satisfies Assumption 4.B with resolution set $[N]$.

   Since $\mathbb{Q}$ is dense in $\mathbb{R}$, there exists a number $\delta(N) < 1/(4N)$ such that $1/(4N) + \delta(N)$ is an irrational number. Let $G'''_N = ([0, 1], \mathcal{F}, \lambda, E'''_N)$ where

   $$E'''_N := \left\{ \left( x, x \pm \left( \frac{1}{4N} + \delta(N) \right) \mod 1 \right) : x \in [0, 1] \right\} \tag{33}$$

   That $G'''_N$ is locally equivalent to $G$ is easily seen via the ambiguity of selecting $a$ when defining $G$. Now we show that $A(G'''_N)$ satisfies Assumption 4.B with resolution set $[N]$. Pick $M \leq N$ and $f \in \mathcal{F}_{\text{reg}(C_v)}$ that is a.e. $C_v$-Lipschitz on $[0, 1]$. Set $\epsilon = \frac{1}{4N} + \delta(N)$ and let:

   $$E = [0, \epsilon) \cup [1 - \epsilon, 1). \tag{34}$$

   Then $\lambda(E) = 2\epsilon < \frac{1}{N}$. Pick $x, x' \in [0, 1] \setminus E$, then $x \pm \epsilon$ and $x' \pm \epsilon$ do not 'loop over' in the interval $[0, 1]$ ( $y \mod 1 = y$, for $y \in \{x, x'\} + \{\pm \epsilon\}$ ). Thus we have:

   $$|Af(x) - Af(x')| = \frac{|f(x + \epsilon) + f(x - \epsilon) - f(x' + \epsilon) - f(x' - \epsilon)|}{2} \tag{35}$$

   $$\leq \frac{|f(x + \epsilon) - f(x' + \epsilon)| + |f(x - \epsilon) - f(x' - \epsilon)|}{2} \tag{36}$$

   $$\leq C_v |x - x'|, \tag{37}$$

   where in the last line we use Lipschitz property of $f$. This finishes the proof.

   $\square$

**Lemma 6** (Well-behaved operators - General graphs). *Let $G$ be a (potentially countably infinite) graph with a coloring $C : V(G) \to [N]$ for some $N$ such that for each vertex $u, v$ with the same color, the multisets of their neighbors' colors $\{C(u') : (u', u) \in E\}$ are the same. Additionally, assume that the cardinality of vertices of each color is the same: that there is a bijection from $\{v : C(v) = c\}$ to $\{v : C(v) = c;\}$ for any colors $c, c'$. Then its adjacency operator satisfies Assumption 3.A with resolution $N$.*

*Proof.* Given that the cardinality of vertices of each color is the same, it is straightforward to map (via a Lebsesgue-measure preserving bijection) vertices of $V$ into equipartition of $[0, 1]$ into $N$ pieces $I_1, ..., I_N$ such that each partition contains vertices of the same color and vertices from different partitions will have different colors. Let $A$ be the normalized adjacency operator of $G$ and $f$ be a function with finite $L^2$ norm such that $f$ is constant on each $I_j$ for each $j$ from 1 to $N$. The goal is to show that $Af$ is also constant on these pieces.

We show this by direct computation. Pick a vertex $x$ and another vertex $y$ from the same piece, say $I_k$ for some particular $k$. By our construction, $x$ and $y$ have the same color since they come from the same piece. By our assumption, the multisets of their neighbors' colors is the same. However, since $f$ is constant on each pieces, we have $f(z) = f(u)$ for $u, v$ of the same color. Therefore, the multisets $\{f(z) : (z, x) \in E\}$ and $\{f(z) : (z, y) \in E\}$ is exactly the same. Taking the appropriate countable sum over each multiset thus result in the same number, i.e. $Af(x) = Af(y)$, which finishes the proof. $\square$

Note that we can drop the countable requirement of the previous proof by invoking an appropriate integral definition. Now we show that the hypercubes - an intermediate graph that is neither dense nor bounded degree, satisfies the requirements of Lemma 6.

**Lemma 7** (Well-behaved operators - Polymer graphs). *Let $P$ be a polymer graphing on finite monomer $n$-vertex graph $M = (V_M, E_M)$. Let $A$ be then normalized (by each vertex degree) adjacency operator of a graphing. For each $N \in \mathbb{N}$, there exists locally equivalent graphings:*

1. *$P_N'$ such that $A(P_n')$ satisfies Assumption 3.A with resolution set $nd(N) = \{nx \in \mathbb{N} : x\alpha = N \text{ for some } \alpha \in \mathbb{N}\}$.*

2. *$P_N''$ such that $A(P_N'')$ satisfies Assumption 4.A with resolution set $n[N]$.*

3. *$P_N'''$ such that $A(P_N''')$ satisfies Assumption 4.B with resolution set $n[N]$.*

*Proof.* The exact same technique as Lemma 5 (make copies of 'shrinked down' graphings to get the result with Assumption 3.A and find a small enough $\delta(N)$ so that the wrap around has small measure to get the result with Assumption 4.A and Assumption 4.B) works. Intuitively, this is because we are replacing each node in the path graphing with a monomer graph.

To handle coloring at the monomers details, use Lemma 6 with the coloring that assigns the same color to the same node in each monomers (since each monomer is a copy of each other, there is an isomorphism between the nodes and by 'the same node' we meant under this isomorphism). $\square$

**Lemma 8** (Well-behaved operators - Graphons). *Let $W : [0, 1]^2 \to [0, 1]$ be a Lipschitz graphon, with Lipschitz constant $C$. In other words, $W$ has finite $L^2$ norm and is Lipschitz in both variables. Then the Hilbert-Schmidt integral operator $H$ that defines the adjacency operator of $W$ satisfies Assumption 3.B at any resolution when applied to graph(on) signal $f$ with $L^1$ norm 1.*

*Proof.* The proof follows from definition. Take $f \in L^2([0,1])$ such that $\|f\|_{L^2} = 1$, then:

$$|Hf(x) - Hf(y)| = \left| \int_0^1 W(x,z)f(z)\mathrm{d}z - \int_0^1 W(y,z)f(z)\mathrm{d}z \right| \tag{38}$$

$$\leq \int_0^1 |W(x,z) - W(y,z))| \cdot |f(z)|\mathrm{d}z \tag{39}$$

$$\leq \int_0^1 C|x-y| \cdot |f(z)|\mathrm{d}z = C|x-y|, \tag{40}$$

where the first line is definition of the Hilbert-Schmidt operator $H$, the second line is triangle inequality and the last line is due to Lipschitzness of $W$ and $L^1$ norm of $f$ □

**Lemma 9** (Well-behaved operators - Hypercubes). *Let $C$ be a hypercube of dimension $N$. Then the normalized adjacency matrix of $C$ satisfies assumption 3.A with resolutions $2^{[n]}$ for each $n < N$.*

*Proof.* We first display a mapping of the vertices in the hypercube over the interval $[0,1]$ such that Assumption 3.A will be shown to hold. Recall that a hypercube vertices can be represented by a binary string of $N$ numbers. Here, similarly, we will associate a vertex $v$ of a hypercube with a number in $[0,1]$ by adding a 0. in front of its binary representation. For example, 0.110101 (as a binary number) is a vertex correspond to the string 110101 when $N = 6$. We call this representation mapping $f : V \to [0,1]$. Two vertices $u, v$ are connected by an edge iff $f(v)$ and $f(v)$ differs by exactly one digit in their binary representation.

To get a hypercube representation over the interval $[0,1]$, we simply take disconnected copies of (uncountably) infinitely many hypercubes described above. To be more precise, for a number $x \in [0,1]$, let $A(x)$ be the $N$-letter binary string such that $0.A$ (as a string) is the trunction of $x$ to the $N$-th digit after the binary point. Then, $x$ is connected to $y \in [0,1]$ iff $A(x)$ and $A(y)$ differs in exactly one digit and $x - A(x) = y - A(y)$. For example, for $N = 4$, we connect 0.110101 with 0.010101 , 0.100101, 0.111101 and 0.110001. Notice how the first four digits after the dots differ from the original number at exactly one letter, and the last few digits always stay the same. In this example, $A(x) = 0.1101$ for $x = 0.110101$.

With this vertex representation in mind, we verify the conditions of Lemma 6. Let $V$. Fix $n < N$ and divide $[0,1]$ into $2^n$ equipartitions and let vertices in the same partition enjoy the same color. We will also number the partition/color by the binary representation of their left hand endpoint. For example, the first interval is labelled 00..0 ($n$ digits), the second interval is labelled 00..01 ($n$ digits). Select a vertex $x$ from a partition with label $a_1 a_2 \ldots a_n$. Then, by the definition of our hypercube representation, the neighbors of $x$ have colors:

- Exactly 1 neighbor with color $(a_1 + 1 \mod 1)a_2 \ldots a_n$.

- Exactly 1 neighbor with color $a_1(a_2 + 1 \mod 1)a_3 \ldots a_n$.

- $\ldots$

- Exactly 1 neighbor with color $a_1 a_2 \ldots a_{n-1}(a_n + 1 \mod 1)$.

- Remaining $N - n$ neighbors all have color $a_1 a_2 \ldots a_n$.

Therefore, the multiset of neighbors' colors of $x$ only depends on the fact that $x$ comes from a partition with label $a_1 \ldots a_n$ and thus, two vertices of the same color has the same multiset of neighbor's colors. Since the condition of Lemma 6 holds, we have the conclusion for this particular resolution $n$. Since $n$ was chosen arbitrarily, the result holds for every $n < N$. □

In future work, we formalizes the way in which the above proof holds for limiting object of hypercube.

## D.2 Proof of Theorem 2

In this section, we prove a slightly more general version of Theorem 2.

**Theorem 5** (General approximation theorem). *Let $A : \mathcal{F} \to \mathcal{F}$ be a P-operator satisfying Assumption 2 with constant $C_A$; Assumption 5.A with constant $C_c$ and resolutions in $\mathcal{N}$. Fix $n \in \mathcal{N}$ and consider $(k, C_v)$-profiles. Let $A_n : \mathcal{F}_n \to \mathcal{F}_n$ be a discretization of $A$ as defined in Equation (9). Then:*

$$d_M(A, A_n) \leq 8\left(\sqrt{\frac{C_A C_v}{n}} + \frac{C_v + C_c}{n}\right). \tag{41}$$

*If instead of Assumption 5.A, A satisfies Assumption 5.B with constant $C_c$ and resolution set $\mathcal{N}$, then:*

$$d_M(A, A_n) \leq 8\left(\sqrt{\frac{C_A C_v + 1}{n}} + \frac{C_v + C_c + 1}{n}\right). \tag{42}$$

*If instead of Assumption 5.A and 5.B, A satisfies Assumption 4.B with resolution set $\mathcal{N}$ then:*

$$d_M(A, A_n) \leq 8\left(\sqrt{\frac{C_A C_v + 1}{n}} + \frac{C_v + 1}{n}\right). \tag{43}$$

*Proof of Theorem 2.* Fix $k \in \mathbb{N}$. In order to derive an upper bound for $d_M$, we find an upper bound for the Hausdorff distance $d_H$ between the two $k$-profiles of $A$ and $A_n$.

**Bounding** $\sup_{\eta \in \mathcal{S}_{k,C_v}(A)} \inf_{\eta_n \in \mathcal{S}_{k,C_v}(A_n)} d_{LP}(\eta, \eta_n)$. To bound the sup inf quantity, we first select an arbitrary $\overline{\eta} \in \mathcal{S}_{k,C_v}(A)$. From this measure, we will construct a measure $\overline{\eta}_n \in \mathcal{S}_{k,C_v}(A_n)$. If we can upper bound $d_{LP}(\overline{\eta}, \overline{\eta}_n) < M$ then we have:

$$\inf_{\eta_n \in \mathcal{S}_{k,C_v}(A_n)} d_{LP}(\overline{\eta}, \eta_n) \leq d_{LP}(\overline{\eta}, \overline{\eta}_n) < M, \text{ for all } \overline{\eta} \in \mathcal{S}_{k,C_v}(A). \tag{44}$$

If further $M$ does not depend on the choice of $\overline{\eta}$, then we have

$$\sup_{\eta \in \mathcal{S}_{k,C_v}(A)} \inf_{\eta_n \in \mathcal{S}_{k,C_v}(A_n)} d_{LP}(\eta, \eta_n) \leq M.$$

We now proceed with this plan. Fix an arbitrary $\overline{\eta} \in \mathcal{S}_k(A)$, by definition of $k$-profiles, there is a corresponding tuple $F = (f_1, \ldots, f_k)$ with elements in $\mathcal{F}_{\mathrm{reg}(C_v)}$ such that $\mathcal{D}_A(F) = \overline{\eta}$.

Form:

$$F' := \left\{ \mathcal{F}_{n,\mathrm{reg}(C_v)} \ni f_j' : u \mapsto n \int_{u-1/n}^{u} f_j \mathrm{d}\lambda \mid j \in [k] \right\}, \qquad \overline{\eta}_n := \mathcal{D}_{A_n}(F'). \tag{45}$$

That $f_j' \in \mathcal{F}_{n,\mathrm{reg}(C_v)}$ is asserted in Lemma 10.

**Bounding** $d_{LP}(\overline{\eta}, \overline{\eta}_n)$ For some $\epsilon > 0$ to be we want to show that $d_{LP}(\overline{\eta}, \overline{\eta}_n) \leq \epsilon$, which is equivalent to showing, for any $U \in \mathcal{B}_{2k}$,

$$\overline{\eta}(U) \leq \overline{\eta}_n(U^\epsilon) + \epsilon \qquad \text{and} \qquad \overline{\eta}_n(U) \leq \overline{\eta}(U^\epsilon) + \epsilon. \tag{46}$$

Recall that $U^\epsilon$ was defined in Eqn (18).

Fix any $U \in \mathcal{B}_{2k}$, we have

$$\overline{\eta}(U) = \int_{\mathbb{R}^{2k}} \mathbb{1}_U \mathrm{d}\mathcal{D}_A(F) = \int_0^1 \mathbb{1}_{F_A \in U} \mathrm{d}\lambda, \tag{47}$$

and

$$\overline{\eta}_n(U^\epsilon) = \int_{\mathbb{R}^{2k}} \mathbb{1}_{U^\epsilon} \mathrm{d}\mathcal{D}_{A_n}(F') = \sum_{u \in [n]/n} \frac{1}{n} \mathbb{1}_{F'_{A_n} \in U^\epsilon}. \tag{48}$$

Subtracting both sides yield:

$$\overline{\eta}(U) - \overline{\eta}_n(U^\epsilon) = \sum_{u \in [n]/n} \int_{u-1/n}^{u} \mathbb{1}_{F_A(x) \in U} - \mathbb{1}_{F'_{A_n}(u) \in U^\epsilon} \mathrm{d}\lambda(x). \tag{49}$$

Similarly, we have:

$$\overline{\eta}_n(U) - \overline{\eta}(U^\epsilon) = \sum_{u \in [n]/n} \int_{u-1/n}^{u} \mathbb{1}_{F'_{A_n}(u) \in U} - \mathbb{1}_{F_A(x) \in U^\epsilon} \mathrm{d}\lambda(x) \tag{50}$$

Let $y_x = \lceil xn \rceil /n$. Recall that $E \subset [0,1]$ defined in Assumption 5.B and 4.B is the set of $x$ where Lipschitzness of the image under $A$ may fail. Let $E = \emptyset$ if we are using Assumption 5.A alternatively. Define the events:

$$\mathcal{E}_U^1(\epsilon) = \{x : F_A(x) \in U \wedge F'_{A_n}(y_x) \notin U^\epsilon\} \tag{51}$$
$$\mathcal{E}_U^2(\epsilon) = \{x : F'_{A_n}(y_x) \in U \wedge F_A \notin U^\epsilon\} \tag{52}$$
$$\mathcal{E}'(\epsilon) = \{x : \|F_A(x) - F'_{A_n}(y_x)\|_2 > \epsilon\} \tag{53}$$
$$\mathcal{E}_j(\epsilon) = \{x : |f_j(x) - f'_j(y_x)| > \epsilon/\sqrt{2k}\}, \qquad j = 1..k \tag{54}$$
$$\mathcal{E}_{j,A}(\epsilon) = \{x : |Af_j(x) - A_n f'_j(y_x)| > \epsilon/\sqrt{2k}\}, \qquad j = 1..k. \tag{55}$$

Using this notation, one also has

$$\overline{\eta}(U) - \overline{\eta}_n(U^\epsilon) \leq \lambda(\mathcal{E}_U^1) \qquad \text{and} \qquad \overline{\eta}_n(U) - \overline{\eta}(U^\epsilon) \leq \lambda(\mathcal{E}_U^2). \tag{56}$$

It is straightforward to see $\mathcal{E}_U^l(\epsilon) \subseteq \mathcal{E}'(\epsilon), l = 1, 2$ for any $U$ by definition of $U^\epsilon$. Furthermore,

$$\mathcal{E}'(\epsilon) \subseteq \bigcup_{j=1}^{k} \mathcal{E}_j(\epsilon) \cup \mathcal{E}_{j,A}(\epsilon), \tag{57}$$

since if all $2k$ dimensions are bounded in absolute value by $\epsilon/\sqrt{2k}$ then the Euclidean distance of the vector is bounded by $\epsilon$.

Therefore it suffices to bound $\lambda(\mathcal{E}_j(\epsilon)) + \lambda(\mathcal{E}_{j,A}(\epsilon))$ for each $j \in [k]$.

**Bounding $\lambda(\mathcal{E}_j(\epsilon))$.** Since $f_j$ is $C_v$-Lipschitz for all $j$, we have:

$$|f_j(x) - f'_j(y_x)| = \left| f_j(x) - n \int_{y_x-1/n}^{y_x} f_j(z) \mathrm{d}\lambda(z) \right| \tag{58}$$

$$= n \left| \int_{y_x-1/n}^{y_x} f_j(x) - f_j(z) \mathrm{d}\lambda(z) \right| \tag{59}$$

$$= n \int_{y_x-1/n}^{y_x} |f_j(x) - f_j(z)| \mathrm{d}\lambda(z) \tag{60}$$

$$\leq n \int_{y_x-1/n}^{y_x} \frac{C_v}{n} \mathrm{d}\lambda(z) = \frac{C_v}{n}, \tag{61}$$

where the first line is definition of $f'_j$, the second line is because $\lambda((u-1/n, u]) = \frac{1}{n}$, the third line is by triangle inequality and the last line is because of Lipschitzness of $f_j$.

Thus, choosing $\epsilon > \sqrt{2k} C_v/n$ means that $\lambda(\mathcal{E}_j(\epsilon)) = 0$. (We can tighten this bound by only assuming that $f_j$ is $C_v$-Lipschitz outside a set of small measure.)

**Bounding $\lambda(\mathcal{E}_{j,A})$.** Let $\mathcal{F}_{[-1,1]} \ni \tilde{f}$ be the extension of $f'$ defined as $\tilde{f}(x) = f'(\lceil xn \rceil /n)$ for all $x \in [0,1]$. Note that $\tilde{f}$ is not continuous in general and hence not Lipschitz. We have

for any $x \in [0, 1]$:

$$|Af_j(x) - A_n f'_j(y_x)| = \left| Af_j(x) - n \int_{y_x - \frac{1}{n}}^{y_x} A\tilde{f}_j(z)\mathrm{d}\lambda(z) \right| \tag{62}$$

$$\leq n \int_{y_x - \frac{1}{n}}^{y_x} \left| Af_j(x) - A\tilde{f}_j(z) \right| \mathrm{d}\lambda(z), \tag{63}$$

where we used uniformity of $\lambda$ and triangle inequality. From here, we proceed slightly differently depending on the specific assumptions.

**Proof via Assumption 5.A or 5.B.** If $A$ satisfies Assumption 5.A or Assumption 5.B, then we have for each $x$:

$$|Af_j(x) - A_n f'_j(y_x)| \leq n \int_{y_x - \frac{1}{n}}^{y_x} \left| Af_j(x) - A\tilde{f}_j(x) \right| + \left| A\tilde{f}_j(x) - A\tilde{f}_j(z) \right| \mathrm{d}\lambda(z) \tag{64}$$

$$\leq \left| Af_j(x) - A\tilde{f}_j(x) \right| + n \int_{y_x - \frac{1}{n}}^{y_x} \left| A\tilde{f}_j(x) - A\tilde{f}_j(z) \right| \mathrm{d}\lambda(z). \tag{65}$$

Define the following events:

$$\mathcal{E}^1_{j,A}(\epsilon) := \left\{ x : \left| Af_j(x) - A\tilde{f}_j(x) \right| > \frac{\epsilon}{2\sqrt{2k}} \right\} \tag{66}$$

$$\mathcal{E}^2_{j,A}(\epsilon) := \left\{ x \notin E : n \int_{y_x - 1/n}^{y_x} \left| A\tilde{f}_j(x) - A\tilde{f}_j(z) \right| \mathrm{d}\lambda(z) > \frac{\epsilon}{2\sqrt{2k}} \right\} \tag{67}$$

Then it is clear that $\mathcal{E}_{j,A} \subseteq \mathcal{E}^1_{j,A} \cup \mathcal{E}^2_{j,A} \cup E$ and thus $\lambda(\mathcal{E}_{j,A}) \leq \lambda(\mathcal{E}^1_{j,A}) + \lambda(\mathcal{E}^2_{j,A}) + \lambda(E)$.

**Bounding $\lambda(\mathcal{E}^1_{j,A}(\epsilon))$ via Assumption 5.A or 5.B.** Because of Assumption 2, we have: $\|A\tilde{f}_j - Af_j\|_2 \leq C_A \|\tilde{f}_j - f_j\|_2$. By $L^p$ norms inequality, we have

$$\|A\tilde{f}_j(x) - Af_j(x)\|_1 \leq C_A \|\tilde{f}_j - f_j\|_2 \leq \frac{C_A C_v}{n}. \tag{68}$$

where the last inequality is due to

$$\|\tilde{f}_j - f_j\|_2^2 = \int_0^1 (\tilde{f}_j(x) - f_j(x))^2 \mathrm{d}\lambda(x) \tag{69}$$

$$= \sum_{u \in [n]/n} \int_{u-1/n}^u (f'_j(u/n) - f_j(x))^2 \mathrm{d}\lambda(x) \tag{70}$$

$$= \sum_{u \in [n]/n} \int_{u-1/n}^u \left( n \int_{(u-1)/n}^{u/n} f_j(z)\mathrm{d}\lambda(z) - f_j(x) \right)^2 \mathrm{d}\lambda(x) \tag{71}$$

$$\leq \sum_{u \in [n]/n} \int_{u-1/n}^u n^2 \left( \int_{u-1/n}^u |f_j(z) - f_j(x)|\mathrm{d}\lambda(z) \right)^2 \mathrm{d}\lambda(x) \tag{72}$$

$$\leq \frac{C_v^2}{n^2}. \tag{73}$$

We have:

$$\frac{C_A C_v}{n} \geq \int_0^1 |A\tilde{f}_j(x) - Af_j(x)|dx \tag{74}$$

$$= \int_{\mathcal{E}^1_{j,A}(\epsilon)} |A\tilde{f}_j(x) - Af_j(x)|dx + \int_{[0,1]\setminus\mathcal{E}^1_{j,A}(\epsilon)} |A\tilde{f}_j(x) - Af_j(x)|dx \tag{75}$$

$$\geq \frac{\epsilon}{2\sqrt{2k}} \lambda(\mathcal{E}^1_{j,A}(\epsilon)) + 0. \tag{76}$$

Thus selecting $\epsilon > 2\sqrt{(\sqrt{2k}\sqrt{k}C_A C_v)/n} = 2^{\frac{5}{4}} k^{\frac{3}{4}} (C_A C_v/n)^{\frac{1}{2}}$ gives $\lambda(\mathcal{E}^1_{j,A}(\epsilon)) \leq \frac{\epsilon}{2k}$.

**Bounding $\lambda(\mathcal{E}_{j,A}^2(\epsilon))$ via Assumption 5.A or 5.B.** Notice that $\tilde{f}_j$ is constant in each $(u - 1/n, u]$ and thus by Assumption 5.A or 5.B, $A\tilde{f}_j$ is $C_c$-Lipschitz in each $(u - 1/n, u]\backslash E$. Therefore we have, for $x \in [0, 1]\backslash E$ and $z \in (y_x - 1/n, y_x]\backslash E$:

$$|A\tilde{f}_j(x) - A\tilde{f}_j(z)| \leq C_c|x - z| \leq \frac{C_c}{n}. \tag{77}$$

When $z \in E$, we use the second condition in Assumption 5.B to get $\|A\tilde{f}_j \mathbb{1}_E\|_1 \leq \frac{1}{n}$ and thus for any $x \notin E$:

$$n\int_{y_x-1/n}^{y_x} \left|A\tilde{f}_j(x) - A\tilde{f}_j(z)\right| d\lambda(z) \tag{78}$$

$$= n\int_{(y_x-1/n,y_x]\cap E} \left|A\tilde{f}_j(x) - A\tilde{f}_j(z)\right| d\lambda(z) + n\int_{(y_x-1/n,y_x]\backslash E} \left|A\tilde{f}_j(x) - A\tilde{f}_j(z)\right| d\lambda(z) \tag{79}$$

$$\leq \frac{C_c}{n} + n\int_{(y_x-1/n,y_x]\cap E} \left|A\tilde{f}_j(x)\right| + \left|A\tilde{f}_j(z)\right| d\lambda(z) \tag{80}$$

$$\leq \frac{C_c}{n} + |A\tilde{f}_j(x)|n\lambda(E_x) + \|A\tilde{f}_j \mathbb{1}_{E_x}\|_1 \tag{81}$$

$$\leq \frac{C_c + 1}{n} + |A\tilde{f}_j(x)|n\lambda(E_x), \tag{82}$$

where $E_x = (y_x - 1/n, y_x] \cap E$.

For $x \in \mathcal{E}_{j,A}^2(\epsilon)$, we have:

$$\frac{C_c + 1}{n} + |A\tilde{f}_j(x)|n\lambda(E_x) > \frac{\epsilon}{2\sqrt{2k}}, \tag{83}$$

or equivalently,

$$|A\tilde{f}_j(x)| > \frac{1}{n\lambda(E_x)}\left(\frac{\epsilon}{2\sqrt{2k}} - \frac{C_c + 1}{n}\right) \tag{84}$$

Since $1/n \geq \|A\tilde{f}_j \mathbb{1}_E\|_1$, we have:

$$1 \geq \left(\frac{\epsilon}{2\sqrt{2k}} - \frac{C_c + 1}{n}\right)\int_{\mathcal{E}_{j,A}^2} \frac{1}{\lambda(E_x)} d\lambda(x) + 0 \tag{85}$$

$$\geq \left(\frac{\epsilon}{2\sqrt{2k}} - \frac{C_c + 1}{n}\right) \sum_{u\in[n]/n} \frac{1}{\lambda(E_u)}\int_{\mathcal{E}_{j,A}^2 \cap (u-1/n,u]} 1 d\lambda(x) \tag{86}$$

$$\geq \left(\frac{\epsilon}{2\sqrt{2k}} - \frac{C_c + 1}{n}\right) n\lambda(\mathcal{E}_{j,A}^2). \tag{87}$$

Thus choosing $\epsilon > 4\sqrt{2k}(C_c + 1)/n$ means that:

$$1 \geq \frac{\epsilon}{4\sqrt{2k}} n\lambda(\mathcal{E}_{j,A}^2), \tag{88}$$

or

$$\lambda(\mathcal{E}_{j,A}^2) \leq \frac{4\sqrt{2k}}{\epsilon n}. \tag{89}$$

Finally, choosing $\epsilon > \sqrt{\frac{16k\sqrt{2k}}{n}} = \frac{2^{\frac{9}{4}}k^{\frac{3}{4}}}{n^{\frac{1}{2}}}$ makes $\lambda(\mathcal{E}_{j,A}^2) \leq \frac{\epsilon}{4k}$; and choosing $\epsilon > \frac{4k}{n}$ makes $\lambda(E) < \frac{1}{n} < \frac{\epsilon}{4k}$.

**Putting everything together via Assumption 5.A or 5.B.** Thus, we can choose $\bar{\epsilon} = 8k\left(\sqrt{\frac{C_A C_v + 1}{n}} + \frac{C_v + C_c + 1}{n}\right)$ to get:

$$\lambda(\mathcal{E}'(\epsilon)) \leq \sum_{j=1}^{k} \frac{\bar{\epsilon}}{k} = \bar{\epsilon}, \tag{90}$$

which allows us to conclude:

$$d_{LP}(\bar{\eta}, \bar{\eta}_n) \leq \bar{\epsilon}. \tag{91}$$

Since $\bar{\eta}$ was chosen arbitrarily, we have for all $\bar{\eta} \in \mathcal{S}_k(A)$,

$$\inf_{\eta_n \in \mathcal{S}_{k,C_v}(A_n)} d_{LP}(\bar{\eta}, \eta_n) \leq d_{LP}(\bar{\eta}, \bar{\eta}_n) \leq \bar{\epsilon}. \tag{92}$$

Thus we also have:

$$\sup_{\eta \in \mathcal{S}_{k,C_v}(A)} \inf_{\eta_n \in \mathcal{S}_{k,C_v}(A_n)} d_{LP}(\eta, \eta_n) \leq \bar{\epsilon}. \tag{93}$$

**Proof via Assumption 4.B.** Here, we use the other triangle inequality to get for each $x$:

$$|Af_j(x) - A_n f_j'(y_x)| \leq n \int_{y_x - \frac{1}{n}}^{y_x} |Af_j(x) - Af_j(z)| + \left|Af_j(z) - A\tilde{f}_j(z)\right| d\lambda(z) \tag{94}$$

$$\leq n \int_{y_x - \frac{1}{n}}^{y_x} |Af_j(x) - Af_j(z)| \, d\lambda(z) + n \int_{y_x - \frac{1}{n}}^{y_x} \left|Af_j(z) - A\tilde{f}_j(z)\right| d\lambda(z). \tag{95}$$

Define the following events:

$$\mathcal{E}_{j,A}^1(\epsilon) := \left\{ x : n \int_{y_x - \frac{1}{n}}^{y_x} \left|Af_j(z) - A\tilde{f}_j(z)\right| d\lambda(z) > \frac{\epsilon}{2\sqrt{2k}} \right\} \tag{96}$$

$$\mathcal{E}_{j,A}^2(\epsilon) := \left\{ x \notin E : n \int_{y_x - \frac{1}{n}}^{y_x} |Af_j(x) - Af_j(z)| \, d\lambda(z) > \frac{\epsilon}{2\sqrt{2k}} \right\} \tag{97}$$

Then it is clear that $\mathcal{E}_{j,A} \subseteq \mathcal{E}_{j,A}^1 \cup \mathcal{E}_{j,A}^2 \cup E$ and thus $\lambda(\mathcal{E}_{j,A}) \leq \lambda(\mathcal{E}_{j,A}^1) + \lambda(\mathcal{E}_{j,A}^2) + \lambda(E)$.

**Bounding $\lambda(\mathcal{E}_{j,A}^1(\epsilon))$ via Assumption 4.B.** Because of Assumption 2, we have: $\|A\tilde{f}_j - Af_j\|_2 \leq C_A \|\tilde{f}_j - f_j\|_2$. By $L^p$ norms inequality, we have

$$\|A\tilde{f}_j - Af_j\|_1 \leq C_A \|\tilde{f}_j - f_j\|_2 \leq \frac{C_A C_v}{n}. \tag{98}$$

where the last inequality is due to

$$\|\tilde{f}_j - f_j\|_2^2 = \int_0^1 (\tilde{f}_j(x) - f_j(x))^2 d\lambda(x) \tag{99}$$

$$= \sum_{u \in [n]/n} \int_{u-1/n}^{u} (f_j'(u) - f_j(x))^2 d\lambda(x) \tag{100}$$

$$= \sum_{u \in [n]/n} \int_{u-1/n}^{u} \left(n \int_{u-1/n}^{u} f_j(z) d\lambda(z) - f_j(x)\right)^2 d\lambda(x) \tag{101}$$

$$\leq \sum_{u \in [n]/n} \int_{u-1/n}^{u} n^2 \left(\int_{u-1/n}^{u} |f_j(z) - f_j(x)| d\lambda(z)\right)^2 d\lambda(x) \tag{102}$$

$$\leq \frac{C_v^2}{n^2}. \tag{103}$$

Consider:

$$\lambda(\mathcal{E}_{j,A}^1(\epsilon)) \cdot \frac{\epsilon}{2\sqrt{2k}} \leq \int_0^1 n \int_{y_x - \frac{1}{n}}^{y_x} \left| Af_j(z) - A\tilde{f}_j(z) \right| \mathrm{d}\lambda(z)\mathrm{d}\lambda(x) \tag{104}$$

$$= \sum_{u \in [n]/n} \int_{u-1/n}^{u} \left| Af_j(z) - A\tilde{f}_j(z) \right| \mathrm{d}\lambda(z) \tag{105}$$

$$= \|Af_j - A\tilde{f}_j\|_1 \leq \frac{C_A C_v}{n}. \tag{106}$$

Thus selecting $\epsilon > \sqrt{\frac{C_A C_v 4k\sqrt{2k}}{n}} = 2^{\frac{5}{4}} k^{\frac{3}{4}} (C_A C_v/n)^{\frac{1}{2}}$ gives $\lambda(\overline{\mathcal{E}}_{j,A}(\epsilon)) \leq \frac{\epsilon}{2k}$.

**Bounding $\lambda(\mathcal{E}_{j,A}^2(\epsilon))$ via Assumption 4.B.** Notice that $f_j$ is Lipschitz in $[0,1]$ and thus by Assumption 4.B, $Af_j$ is $C_v$-Lipschitz in $[0,1]\backslash E$. Therefore we have, for $x \in [0,1]\backslash E$ and $z \in (y_x - 1/n, y_x]\backslash E$ and:

$$|Af_j(x) - Af_j(z)| \leq C_v|x - z| \leq \frac{C_v}{n}. \tag{107}$$

When $z \in E$, we use the second condition in Assumption 4.B to get $\|Af_j \mathbb{1}_E\|_1 \leq \frac{1}{n}$ and thus for any $x \notin E$:

$$n \int_{y_x - 1/n}^{y_x} |Af_j(x) - Af_j(z)| \, \mathrm{d}\lambda(z) \tag{108}$$

$$= n \int_{(y_x - 1/n, y_x] \cap E} |Af_j(x) - Af_j(z)| \, \mathrm{d}\lambda(z) + n \int_{(y_x - 1/n, y_x]\backslash E} |Af_j(x) - Af_j(z)| \, \mathrm{d}\lambda(z) \tag{109}$$

$$\leq \frac{C_v}{n} + n \int_{(y_x - 1/n, y_x] \cap E} |Af_j(x)| + |Af_j(z)| \, \mathrm{d}\lambda(z) \tag{110}$$

$$\leq \frac{C_v}{n} + |Af_j(x)| n\lambda(E_x) + \|Af_j \mathbb{1}_{E_x}\|_1 \tag{111}$$

$$\leq \frac{C_v + 1}{n} + |Af_j(x)| n\lambda(E_x), \tag{112}$$

where $E_x = (y_x - 1/n, y_x] \cap E$.

For $x \in \mathcal{E}_{j,A}^2(\epsilon)$, we have:

$$\frac{C_v + 1}{n} + |Af_j(x)| n\lambda(E_x) > \frac{\epsilon}{2\sqrt{2k}}, \tag{113}$$

or equivalently,

$$|Af_j(x)| > \frac{1}{n\lambda(E_x)} \left( \frac{\epsilon}{2\sqrt{2k}} - \frac{C_v + 1}{n} \right) \tag{114}$$

Since $1/n \geq \|Af_j \mathbb{1}_E\|_1$, we have:

$$1 \geq \left( \frac{\epsilon}{2\sqrt{2k}} - \frac{C_v + 1}{n} \right) \int_{\mathcal{E}_{j,A}^2} \frac{1}{\lambda(E_x)} \mathrm{d}\lambda(x) + 0 \tag{115}$$

$$\geq \left( \frac{\epsilon}{2\sqrt{2k}} - \frac{C_v + 1}{n} \right) \sum_{u \in [n]/n} \frac{1}{\lambda(E_u)} \int_{\mathcal{E}_{j,A}^2 \cap (u-1/n, u]} 1 \mathrm{d}\lambda(x) \tag{116}$$

$$\geq \left( \frac{\epsilon}{2\sqrt{2k}} - \frac{C_v + 1}{n} \right) n\lambda(\mathcal{E}_{j,A}^2). \tag{117}$$

Thus choosing $\epsilon > 4\sqrt{2k}(C_v + 1)/n$ means that:

$$1 \geq \frac{\epsilon}{4\sqrt{2k}} n\lambda(\mathcal{E}_{j,A}^2), \tag{118}$$

or

$$\lambda(\mathcal{E}_{j,A}^2) \leq \frac{4\sqrt{2k}}{\epsilon n}. \tag{119}$$

Finally, choosing $\epsilon > \sqrt{\frac{16k\sqrt{2k}}{n}} = \frac{2^{\frac{9}{4}}k^{\frac{3}{4}}}{n^{\frac{1}{2}}}$ makes $\lambda(\mathcal{E}_{j,A}^2) \leq \frac{\epsilon}{4k}$; and choosing $\epsilon > \frac{4k}{n}$ makes $\lambda(E) < \frac{1}{n} < \frac{\epsilon}{4k}$.

**Putting everything together via Assumption 4.B.** Thus, we can choose $\overline{\epsilon} = 8k\left(\sqrt{\frac{C_A C_v + 1}{n}} + \frac{C_v + 1}{n}\right)$ to get:

$$\lambda(\mathcal{E}'(\epsilon)) \leq \sum_{j=1}^{k} \frac{\overline{\epsilon}}{k} = \overline{\epsilon}, \tag{120}$$

which allows us to conclude:

$$d_{LP}(\overline{\eta}, \overline{\eta}_n) \leq \overline{\epsilon}. \tag{121}$$

Since $\overline{\eta}$ was chosen arbitrarily, we have for all $\overline{\eta} \in \mathcal{S}_k(A)$,

$$\inf_{\eta_n \in \mathcal{S}_{k,C_v}(A_n)} d_{LP}(\overline{\eta}, \eta_n) \leq d_{LP}(\overline{\eta}, \overline{\eta}_n) \leq \overline{\epsilon}. \tag{122}$$

Thus we also have:

$$\sup_{\eta \in \mathcal{S}_{k,C_v}(A)} \inf_{\eta_n \in \mathcal{S}_{k,C_v}(A_n)} d_{LP}(\eta, \eta_n) \leq \overline{\epsilon}. \tag{123}$$

**Bounding** $\sup_{\eta_n \in \mathcal{S}_{k,C_v}(A_n)} \inf_{\eta \in \mathcal{S}_{k,C_v}(A)} d_{LP}(\eta, \eta_n)$. In this direction, we proceed identically, but now choose $\overline{\eta}_n$ arbitrarily in $\mathcal{S}_{k,C_v}(A_n)$. By definition of $(k, C_v)$-profiles, there exists a tuple $F = (f_1, \ldots, f_k)$ each in $\mathcal{F}_{n,\mathrm{reg}(C_v)}$ such that $\overline{\eta}_n = \mathcal{D}_{A_n}(F)$.

Construct

$$F' := \left\{\mathcal{F}_{\mathrm{reg}(C_v)} \ni f_j' : x \mapsto (1 - y_x n + xn)f_j(y_x) + (y_x n - xn)f_j(y_x - 1/n)\right\}, \tag{124}$$
$$\overline{\eta} = \mathcal{D}_A(F'), \tag{125}$$

where $y_x = \lceil xn \rceil / n$ for each $j \in [k]$. That $f_j' \in \mathcal{F}_{\mathrm{reg}(C_v)}$ is asserted in Lemma 11. Intuitively, $f_j'$ is the continuous piecewise linear function that interpolates $f_j$, created by joining $f_j(u - 1/n)$ and $f_j(u)$ with a line segment for each $u \in [n]/n$. Note also that $f_j'(u) = f_j(u)$ for all $u \in [n]/n$.

**Bounding** $d_{LP}(\overline{\eta}, \overline{\eta}_n)$. As with the previous direction, we start with an arbitrary $U \in \mathcal{B}_k$ and write down the differences:

$$\overline{\eta}_n(U) - \overline{\eta}(U^\epsilon) = \sum_{u \in [n]/n} \int_{u-1/n}^{u} \mathbb{1}_{F_{A_n}(u) \in U} - \mathbb{1}_{F_A'(x) \in U^\epsilon} \mathrm{d}\lambda(x), \tag{126}$$

$$\overline{\eta}(U) - \overline{\eta}_n(U^\epsilon) = \sum_{u \in [n]/n} \int_{u-1/n}^{u} \mathbb{1}_{F_A'(x) \in U} - \mathbb{1}_{F_{A_n}(u) \in U^\epsilon} \mathrm{d}\lambda(x). \tag{127}$$

Define the events:

$$\mathcal{E}_U^1(\epsilon) = \{x : F_{A_n}(y_x) \in U \wedge F_A'(x) \notin U^\epsilon\} \tag{128}$$
$$\mathcal{E}_U^2(\epsilon) = \{x : F_A'(x) \in U \wedge F_{A_n}(y_x) \notin U^\epsilon\} \tag{129}$$
$$\mathcal{E}'(\epsilon) = \{x : \|F_{A_n}(y_x) - F_A'(x)\|_2 > \epsilon\} \tag{130}$$
$$\mathcal{E}_j(\epsilon) = \{x : |f_j(y_x) - f_j'(x)| > \epsilon/\sqrt{2k}\}, \qquad j = 1..k \tag{131}$$
$$\mathcal{E}_{j,A}(\epsilon) = \{x : |A_n f_j(y_x) - A f_j'(x)| > \epsilon/\sqrt{2k}\}, \qquad j = 1..k. \tag{132}$$

Using this notation, one also has

$$\bar{\eta}(U) - \bar{\eta}_n(U^\epsilon) \le \lambda(\mathcal{E}_U^1) \qquad \text{and} \qquad \bar{\eta}_n(U) - \bar{\eta}(U^\epsilon) \le \lambda(\mathcal{E}_U^2). \tag{133}$$

It is straightforward to see $\mathcal{E}_U^l(\epsilon) \subseteq \mathcal{E}'(\epsilon), l = 1, 2$ for any $U$ by definition of $U^\epsilon$. Furthermore,

$$\mathcal{E}'(\epsilon) \subseteq \bigcup_{j=1}^{k} \mathcal{E}_j(\epsilon) \cup \mathcal{E}_{j,A}(\epsilon), \tag{134}$$

since if all $2k$ dimensions are bounded in absolute value by $\epsilon/\sqrt{2k}$ then the Euclidean distance of the vector is bounded by $\epsilon$.

Therefore it suffices to bound $\lambda(\mathcal{E}_j(\epsilon)) + \lambda(\mathcal{E}_{j,A}(\epsilon))$ for each $j \in [k]$.

**Bounding $\lambda(\mathcal{E}_j(\epsilon))$.** Since $f_j'$ is $C_v$-Lipschitz for all $j$ (Lemma 11), we have:

$$|f_j(y_x) - f_j'(x)| = |f_j'(y_x) - f_j'(x)| \le \frac{C_v}{n}. \tag{135}$$

Thus, choosing $\epsilon > \sqrt{2k}C_v/n$ means that $\lambda(\mathcal{E}_j(\epsilon)) = 0$. (We can tighten this bound by only assuming that $f_j$ is $C_v$-Lipschitz outside a set of small measure.)

**Bounding $\lambda(\mathcal{E}_{j,A})$.** Let $\mathcal{F}_{[-1,1]} \ni \tilde{f}$ be the extension of $f$ defined as $\tilde{f}(x) = f(\lceil xn \rceil /n)$ for all $x \in [0,1]$. Note that $\tilde{f}$ is not continuous in general and hence not Lipschitz. We have for any $x \in [0,1]$:

$$|Af_j'(x) - A_n f_j(y_x)| = \left| Af_j'(x) - n \int_{y_x - \frac{1}{n}}^{y_x} A\tilde{f}_j(z) d\lambda(z) \right| \tag{136}$$

$$\le n \int_{y_x - \frac{1}{n}}^{y_x} \left| Af_j'(x) - A\tilde{f}_j(z) \right| d\lambda(z), \tag{137}$$

where we used uniformity of $\lambda$ and triangle inequality. The last thing that we need to show is:

$$\|f_j' - \tilde{f}_j\|_2^2 = \int_0^1 ((1 - y_x n + xn)f(y_x) + (xn - y_x n)f(y_x - 1/n) - f(y_x))^2 d\lambda(x) \tag{138}$$

$$= \int_0^1 n^2 (y_x - x)^2 (f(y_x) - f(y_x - 1/n))^2 d\lambda(x) \tag{139}$$

$$\le \frac{C_v^2}{n^2}. \tag{140}$$

From here, by a word-for-word argument, we can show that the same choice of $\bar{\epsilon}$ does the trick to make $\lambda(\mathcal{E}'(\epsilon)) \le \bar{\epsilon}$. This works since we only use the fact that $f_j' \in \mathcal{F}_{\text{reg}(C_v)}$ as well as assumption conditions in the previous proof.

**Bounding $d_M$.** We have:

$$d_H(\mathcal{S}_k(A), \mathcal{S}_k(A_n)) = \max(\sup_{\bar{\eta}_n \in \mathcal{S}_k(A_n)} \inf_{\eta \in \mathcal{S}_k(A)} d_{LP}(\bar{\eta}_n, \eta), \sup_{\bar{\eta} \in \mathcal{S}_k(A)} \inf_{\eta_n \in \mathcal{S}_k(A_n)} d_{LP}(\bar{\eta}, \eta_n)) \le \bar{\epsilon}. \tag{141}$$

Therefore,

$$d_M(A, A_n) \le \sum_{k=1}^{\infty} \frac{8k \left( \sqrt{\frac{C_A C_v + 1}{n}} + \frac{C_v + 1}{n} \right)}{2^k} \le 8 \left( \sqrt{\frac{C_A C_v + 1}{n}} + \frac{2C_c + C_v + 1}{n} \right). \tag{142}$$

$\square$

**Lemma 10.** *Fix $n \in \mathbb{N}$. Fix $f \in \mathcal{F}_{reg(C_v)}$. Define the restriction $f' : \frac{1}{n}[n] \to [-1,1] : u \mapsto n \int_{u-1/n}^{u} f(z) \mathrm{d}\lambda(z)$. Then $f' \in \mathcal{F}_{n,reg(C_v)}$.*

*Proof.* Firstly, we have for any $u \in [n]/n$:

$$|f'(u)| \le n \int_{u-1/n}^{u} |f(z)| \mathrm{d}\lambda(z) \le n \int_{u-1/n}^{u} 1 \mathrm{d}\lambda(z) = 1. \tag{143}$$

Therefore $\|f'\|_{L^2([n]/n)} \le n/n = 1$ and thus $f'$ is measurable. Finally, for any $u < u' \in [n]/n$:

$$|f'(u) - f'(u')| \le n \int_{u-1/n}^{u} |f(z) - f(z + (u'-u))| \mathrm{d}\lambda(z) \le C_v(u'-u), \tag{144}$$

where we use Lipschitz property of $f$ in the last inequality. This shows that $f'$ is also $C_v$-Lipschitz. $\qquad\square$

**Remark 1.** *The restriction to $\mathcal{F}_{[-1,1]}$ is necessary for this lemma to work because $L^2([0,1])$ functions can blow up near 0.*

**Lemma 11.** *Fix $n \in \mathbb{N}$. Fix $f \in \mathcal{F}_{n,reg(C_v)}$. Let $y_x := \lceil xn \rceil /n$ for any $x \in [0,1]$. Define the extension $f' : x \mapsto (1 - y_x n + xn) f(y_x) + (xn - y_x n) f(y_x - 1/n)$. Then $f' \in \mathcal{F}_{reg(C_v)}$.*

*Proof.* Firstly, since $f'$ linearly interpolates between points of $f$, its range cannot exceed that of $f$. The restricted range immediately implies that the $L^2$ norm is bounded by 1 since the support is also in $[0,1]$. Finally, for any $x < x' \in [0,1]$, if there is a $u$ such that $x, x' \in [u - 1/n, u)$ then the fact that the interpolation is linear means that the line segment from $f'(x)$ to $f'(x')$ shares the same slope as that from $f(u)$ to $f(u - 1/n)$, which is at most $C_v$ since $f \in \mathcal{F}_{n,reg(C_v)}$. Otherwise, there exists a $u < u' \in [n]/n$ such that $x \in [u - 1/n, u)$ and $x' \in [u' - 1/n, u')$. We have:

$$|f'(x) - f'(x')| \le |f'(x) - f'(u)| + |f'(u) - f'(u' - 1/n)| + |f'(u' - 1/n) - f'(x')| \le C_v(x' - x), \tag{145}$$

which proves Lipschitzness of $f'$. $\qquad\square$

### D.3 Proofs for Section 4.2

The following result characterizes the behaviors of Assumptions 2, 5.A, 5.B and 4.B under addition, multiplication by scalars, power, and element-wise composition with a 1-Lipschitz map:

**Lemma 12.** *Fix $k \in \mathbb{N}$ and $\alpha \in \mathbb{R}$. Recall that $\rho : \mathbb{R} \to \mathbb{R}$ is a 1-Lipschitz map. Let $A_1, A_2 : \mathcal{F} \to \mathcal{F}$ satisfy Assumption 2 with constant $C_A^1$ and $C_A^2$ respectively; and Assumption 5.A or 5.B or 4.B with constant $C_c^1$ and $C_c^2$ respectively with common resolution set $\mathcal{N}$. Then:*

1. *$A_1 + A_2$ satisfy Assumptions 2 and 5.A or 5.B or 4.B with constant $(C_A^1 + C_A^2)$ and $(C_c^1 + C_c^2)$ respectively.*

2. *$\alpha A_1$ satisfies Assumption 2 and 5.A or 5.B or 4.B with constant $|\alpha| C_A^1$ and $|\alpha| C_c^1$ respectively.*

3. *$\rho A_1$ (where the composition is done element-wise) satisfies Assumption 2 and 5.A or 5.B or 4.B with constant $C_A^1$ and $C_c^1$ respectively.*

4. *Furthermore, if $C_c^1 = 0$ then $A_2 \circ A_1$ satisfies Assumption 2 and 5.A 5.B or 4.B with constant $C_A^1 C_A^2$ and $C_c^2$ respectively.*

*Proof of Lemma 12.* To recall, $A_1, A_2 : \mathcal{F} \to \mathcal{F}$ are $P$-operators that satisfies Assumption 2 with constant $C_A^1$ and $C_A^2$ respectively and Assumption 5.A or 5.B or 4.B with constant $C_c^1$ and $C_c^2$ respectively with common resolution set $\mathcal{N}$. We now show each part of the Lemma.

1. We have, for any $f, g \in \mathcal{F}$:

$$\|(A_1 + A_2)f - (A_1 + A_2)g\|_2 \leq \|A_1 f - A_1 g\|_2 + \|A_2 f - A_2 g\|_2 \qquad (146)$$

$$\leq C_A^1 \|f - g\|_2 + C_A^2 \|f - g\|_2, \qquad (147)$$

where the first line is triangle inequality and the second line is Assumption 2.

Let $f \in \mathcal{F}_{\mathrm{reg}(C_v)}$ piecewise constant on intervals $[n]/n$ and $x, y \in (u - 1/n, u]$ for some $n \in \mathcal{N}$ and $u \in [n]$, we have:

$$|(A_1 + A_2)f(x) - (A_1 + A_2)f(y)| \leq |[A_1 f](x) - [A_1 f](y)| + |[A_2 f](x) - [A_2 f](y)| \qquad (148)$$

$$\leq C_c^1 |x - y| + C_c^2 |x - y|, \qquad (149)$$

where the first line is triangle inequality and the second line is Assumption 5.A.

2. We have, for any $f, g \in \mathcal{F}$:

$$\|(\alpha A_1)f - (\alpha A_1)g\|_2 \leq |\alpha| \cdot \|A_1 f - A_1 g\|_2 \qquad (150)$$

$$\leq |\alpha| C_A^1 \|f - g\|_2, \qquad (151)$$

where the first line is property of norm and the second line is Assumption 2.

Let $f \in \mathcal{F}_{\mathrm{reg}(C_v)}$ piecewise constant on intervals $[n]/n$, and $x, y \in (u - 1/n, u]$ for some $n \in \mathcal{N}$ and $u \in [n]$, we have:

$$|(\alpha A_1)f(x) - (\alpha A_1)f(y)| \leq |\alpha[A_1 f](x) - \alpha[A_1 f](y)| \qquad (152)$$

$$\leq |\alpha| C_c^1 |x - y|, \qquad (153)$$

where the second line is Assumption 5.A.

3. We have, for any $f, g \in \mathcal{F}$:

$$\|(\rho A_1)f - (\rho A_1)g\|_2 \leq \|A_1 f - A_1 g\|_2 \qquad (154)$$

$$\leq C_A^1 \|f - g\|_2, \qquad (155)$$

where the first line is Lipschitz property of $\rho$ and the second line is Assumption 2.

Let $f \in \mathcal{F}_{\mathrm{reg}(C_v)}$ piecewise constant on intervals $[n]/n$ and $x, y \in (u - 1/n, u]$ for some $n \in \mathcal{N}$ and $u \in [n]$, we have:

$$|(\rho A_1)f(x) - (\rho A_1)f(y)| \leq |\rho([A_1 f](x)) - \rho([A_1 f](y))| \qquad (156)$$

$$\leq |A_1 f(x) - A_1 f(y)| \leq C_c^1 |x - y|, \qquad (157)$$

where the second line is Lipschitz property of $\rho$ and Assumption 5.A.

4. We have, for any $f, g \in \mathcal{F}$:

$$\|(A_2 A_1)f - (A_2 A_1)g\|_2 \leq C_A^2 \|A_1 f - A_1 g\|_2 \qquad (158)$$

$$\leq C_A^1 C_A^2 \|f - g\|_2, \qquad (159)$$

where the first line is Lipschitz property of $A_2$ and the second line is that of $A_1$.

Let $f \in \mathcal{F}_{\mathrm{reg}(C_v)}$ piecewise constant on intervals $[n]/n$. Since $A_1$ send constant pieces to constant pieces, the final implication follows direction from Assumption 5.A of $A_2$.

$\square$

**Lemma 13.** *Fix $n \in \mathcal{N}, k \in [K] \cup \{0\}, A : \mathcal{F} \to \mathcal{F}$ satisfies Assumption 2 with constant $C_A$ and Assumption 5.A with constant $C_c$ and resolution set $\mathcal{N}$. Let $n \in \mathcal{N}, f_1 \in \mathcal{F}_n, f_2 \in \mathcal{F}$. Recall that $\tilde{f} \in \mathcal{F}$ denotes the extension of $f \in \mathcal{F}_n$ to $[0, 1]$ as $\tilde{f}(x) = f(\lceil xn \rceil)/n$. If $\|\tilde{f}_1 - f_2\|_2 \leq M$ for some positive constant $M$ then*

$$\|\widetilde{A_n^k f_1} - A^k f_2\|_2^2 \leq \frac{3^{k+1} k C_c^2 C_A^{2k}}{n^2} + 3^k C_A^{2k} M. \qquad (160)$$

*If $A$ satisfies Assumption 5.B with constant $C_c$ instead then the bound becomes:*

$$\|\widetilde{A_n^k f_1} - A^k f_2\|_2^2 \leq \frac{3^{k+1} k (C_c + 1)^2 C_A^{2k}}{n^2} + 3^k C_A^{2k} M. \tag{161}$$

*If $A$ satisfies Assumption 4.B instead then the bound becomes:*

$$\|\widetilde{A_n^k f_1} - A^k f_2\|_2^2 \leq \frac{3^{k+1} k (C_v + 1)^2 C_A^{2k}}{n^2} + 3^k C_A^{2k} M. \tag{162}$$

*Proof.* When $k = 0$, the bound is vacuously true. We will hide the measure in integrals when it is clear from context in this proof.

Assume that the bound is correct up to some $k - 1$. We have:

$$\|\widetilde{A_n^k f_1} - A^k f_2\|_2^2 \tag{163}$$

$$= \sum_{u \in 1/n[n]} \int_{u-\frac{1}{n}}^u (A_n^k f_1(u) - A^k f_2(z))^2 \mathrm{d}z \tag{164}$$

$$= \sum_{u \in 1/n[n]} \int_{u-\frac{1}{n}}^u \left( n \int_{u-\frac{1}{n}}^u A[\widetilde{A_n^{k-1} f_1}](y) \mathrm{d}y - A^k f_2(z) \right)^2 \mathrm{d}z \tag{165}$$

Here we can use Jensen's inequality to get:

$$\|\widetilde{A_n^k f_1} - A^k f_2\|_2^2 \tag{166}$$

$$= n \sum_{u \in 1/n[n]} \int_{u-\frac{1}{n}}^u \int_{u-\frac{1}{n}}^u \left( A[\widetilde{A_n^{k-1} f_1}](y) - A[A^{k-1} f_2](z) \right)^2 \mathrm{d}y \mathrm{d}z \tag{167}$$

$$\tag{168}$$

We proceed slightly differently based on the exact assumptions that we have. If $A$ satisfies Assumption 5.A or 5.B then:

$$\|\widetilde{A_n^k f_1} - A^k f_2\|_2^2 \tag{169}$$

$$\leq n \sum_{u \in 1/n[n]} \int_{u-\frac{1}{n}}^u$$

$$\int_{u-\frac{1}{n}}^u \left( |A[\widetilde{A_n^{k-1} f_1}](y) - A[\widetilde{A_n^{k-1} f_1}](z)| + |A[\widetilde{A_n^{k-1} f_1}](z) - A[A^{k-1} f_2](z)| \right)^2 \mathrm{d}y \mathrm{d}z \tag{170}$$

$$\leq n \sum_{u \in 1/n[n]} \int_{u-\frac{1}{n}}^u$$

$$\int_{u-\frac{1}{n}}^u \left( \frac{C_c + 1}{n} + |A[\widetilde{A_n^{k-1} f_1}](z)| n\lambda(E_z) + |A[\widetilde{A_n^{k-1} f_1}](z) - A[A^{k-1} f_2](z)| \right)^2 \mathrm{d}y \mathrm{d}z \tag{171}$$

$$\leq \int_0^1 \left( \frac{C_c + 1}{n} + |A[\widetilde{A_n^{k-1} f_1}](z)| n\lambda(E_z) + |A[\widetilde{A_n^{k-1} f_1}](z) - A[A^{k-1} f_2](z)| \right)^2 \mathrm{d}z \tag{172}$$

$$\leq \frac{3(C_c + 1)^2}{n^2} + 3 \int_0^1 (A[\widetilde{A_n^{k-1} f_1}](z) n\lambda(E_z))^2 \mathrm{d}z + 3\|A[\widetilde{A_n^{k-1} f_1}] - A[A^{k-1} f_2]\|_2^2 \mathrm{d}z. \tag{173}$$

Here, we give a heuristic argument while the formal argument is exactly similar to that in Theorem 2. The term $3 \int_0^1 (A[\widetilde{A_n^{k-1} f_1}](z) n\lambda(E_z))^2 \mathrm{d}z$ can be made as close to 0 as possible simply by changing the requirement on $\lambda(E)$ to be smaller and smaller (see the discussion

after Assumption 4.A). At the same time, our assumptions ensure that since $\widetilde{A_n^{k-1}f_1}$ is a piecewise constant function, the action of $A$ on it cannot become too wild (Lipschitz outside of $E$ and bounded $L^1$ norm inside $E$). This heuristic argument, when made rigorous can help us conclude that:

$$\|\widetilde{A_n^k f_1} - A^k f_2\|_2^2 \le \frac{3(C_c+1)^2}{n^2} + 3C_A^2\|\widetilde{A_n^{k-1}f_1} - A^{k-1}f_2\|_2^2 \tag{174}$$

Plug in the inductive hypothesis and solve the recurrent to get the bound.

Similarly, if we are instead using Assumption 4.B then we use the other triangle inequality to conclude:

$$\|\widetilde{A_n^k f_1} - A^k f_2\|_2^2 \tag{175}$$

$$\le n \sum_{u \in 1/n[n]} \int_{u-\frac{1}{n}}^{u}$$

$$\int_{u-\frac{1}{n}}^{u} \left( |A[\widetilde{A_n^{k-1}f_1}](y) - A[A^{k-1}f_1](y)| + |A[A^{k-1}f_1](y) - A[A^{k-1}f_2](z)| \right)^2 \mathrm{d}y\mathrm{d}z \tag{176}$$

$$\le 2n \sum_{u \in 1/n[n]} \int_{u-\frac{1}{n}}^{u} \int_{u-\frac{1}{n}}^{u} \left( A[\widetilde{A_n^{k-1}f_1}](y) - A[A^{k-1}f_1](y) \right)^2 \mathrm{d}y\mathrm{d}z$$

$$+ 2n \sum_{u \in 1/n[n]} \int_{u-\frac{1}{n}}^{u} \int_{u-\frac{1}{n}}^{u} \left( \frac{C_v+1}{n} + |A[A^{k-1}f_2](z)|n\lambda(E_z) \right)^2 \mathrm{d}y\mathrm{d}z \tag{177}$$

$$\le \frac{3(C_v+1)^2}{n^2} + 3\int_0^1 (A[A^{k-1}f_1](z)n\lambda(E_z))^2 \mathrm{d}z + 3\|A[\widetilde{A_n^{k-1}f_1}] - A[A^{k-1}f_2]\|_2^2. \tag{178}$$

Again, we will make use of a heuristic argument to argue that the middle term can be controlled by controlling $\lambda(E)$ since $A$ sends Lipschitz functions to Lipschitz functions outside of $E$ and has bounded $L^1$ norm inside of $E$. Therefore:

$$\|\widetilde{A_n^k f_1} - A^k f_2\|_2^2 \le \frac{3(C_v+1)^2}{n^2} + 3\|A[\widetilde{A_n^{k-1}f_1}] - A[A^{k-1}f_2]\|_2^2. \tag{179}$$

Solve the recurrent to get the result in the statement of the Lemma. $\square$

**Lemma 14.** *In the same setting as Lemma 13, recall that a tilde over a function in $\mathcal{F}_n$ denotes its extension to $\mathcal{F}$. If $A$ satisfies Assumption 2 with constant $C_A$ and Assumption 5.A with constant $C_c$ and resolution set $\mathcal{N}$. Let $\Phi(h,A,\cdot) = \rho\left(\sum_{g \in n_l}\sum_{k=0}^{K-1} A^k \Phi_g(h,A,\cdot)\right)$ for some $\Phi_g$ graphop neural network such that $\|\widetilde{\Phi_g}(h,A_n,f) - \Phi_g(h,A,\tilde{f})\|_2 < M$ for all $g \in [n_l]$. Then:*

$$\|\widetilde{\Phi}(h,A_n,f) - \Phi(h,A,\tilde{f})\|_2^2 \le K^2 n_l^2 3^K \left( Kn^{-2}C_c^2 C_A^{2K} + C_A^{2K}M \right). \tag{180}$$

*If instead $A$ satisfies Assumption 5.B with constant $C_c$ then the bound becomes:*

$$\|\widetilde{\Phi}(h,A_n,f) - \Phi(h,A,\tilde{f})\|_2^2 \le K^2 n_l^2 3^K \left( Kn^{-2}(C_c+1)^2 C_A^{2K} + C_A^{2K}M \right). \tag{181}$$

*If instead $A$ satisfies Assumption 4.B then the bound becomes:*

$$\|\widetilde{\Phi}(h,A_n,f) - \Phi(h,A,\tilde{f})\|_2^2 \le K^2 n_l^2 3^K \left( Kn^{-2}(C_v+1)^2 C_A^{2K} + C_A^{2K}M \right). \tag{182}$$

*Proof.* We have:

$$\|\widetilde{\Phi}(h, A_n, f) - \Phi(h, A, \tilde{f})\|_2^2 \tag{183}$$

$$= \sum_{u \in \frac{1}{n}[n]} \int_{u-1/n}^{u} \left( \rho\left( \sum_g \sum_k \widetilde{A_n^k \Phi_g}(h, A_n, f)(x) \right) - \rho\left( \sum_g \sum_k A^k \Phi_g(h, A, \tilde{f})(x) \right) \right)^2 \mathrm{d}\lambda(x) \tag{184}$$

$$\leq \sum_{u \in \frac{1}{n}[n]} \int_{u-1/n}^{u} \left( \sum_g \sum_k \widetilde{A_n^k \Phi_g}(h, A_n, f)(x) - A^k \Phi_g(h, A, \tilde{f})(x) \right)^2 \mathrm{d}\lambda(x) \tag{185}$$

$$\leq \sum_{u \in \frac{1}{n}[n]} \int_{u-1/n}^{u} \left( \sum_g \sum_k \left| \widetilde{A_n^k \Phi_g}(h, A_n, f)(x) - A^k \Phi_g(h, A, \tilde{f})(x) \right| \right)^2 \mathrm{d}\lambda(x) \tag{186}$$

$$\leq K n_l \sum_{u \in \frac{1}{n}[n]} \int_{u-1/n}^{u} \sum_g \sum_k \left( \left| \widetilde{A_n^k \Phi_g}(h, A_n, f)(x) - A^k \Phi_g(h, A, \tilde{f})(x) \right| \right)^2 \mathrm{d}\lambda(x) \tag{187}$$

$$= K n_l \sum_g \sum_k \| \widetilde{A_n^k \Phi_g}(h, A_n, f)(x) - A^k \Phi_g(h, A, \tilde{f})(x) \|_2^2 \tag{188}$$

$$\leq K n_l \sum_g \sum_k \frac{3^{k+1} k C_c^2 C_A^{2k}}{n^2} + 3^k C_A^{2k} \| \widetilde{\Phi}_g(h, A_n, f) - \Phi_g(h, A, \tilde{f}) \|_2^2 \tag{189}$$

$$\leq n^{-2} n_L^2 \cdot K^3 \cdot C_c^2 \cdot 3^K C_A^{2K} + K^2 \cdot n_l^2 \cdot 3^K C_A^{2K} \cdot M \tag{190}$$

$$= K^2 n_l^2 3^K \left( K n^{-2} C_c^2 C_A^{2K} + C_A^{2K} M \right), \tag{191}$$

where in the first line, we use the fact that continuous extension commutes with finite element-wise sum and element-wise application of $\rho$, while expanding $L^2$ norm; the second line uses Lipschitz property of $\rho$; the third line uses triangle inequality; the fourth line uses equivalence of $p$-norms in finite dimensional vectors; the fifth line is again $L^2$ norm definition; the sixth line applies Lemma 13 and the rest is algebra.

To get the rest of the cases, applies different versions of Lemma 13. $\qquad\square$

**Lemma 15.** *Let $\Phi$ be an $L$-layer graphop neural network in the same setting as Lemma 13. If $A$ satisfies Assumption 5.A with constant $C_c$ then:*

$$\|\widetilde{\Phi}(h, A_n, f) - \Phi(h, A, \tilde{f})\|_2 \leq n^{-1}(3^K K n_{\max} C_A^K)^L \max\left( C_v, 3^K K^2 C_c n_{\max} C_A^K \right). \tag{192}$$

*If instead $A$ satisfies Assumption 5.B with constant $C_c$ then the bound becomes:*

$$\|\widetilde{\Phi}(h, A_n, f) - \Phi(h, A, \tilde{f})\|_2 \leq n^{-1}(3^K K n_{\max} C_A^K)^L \max\left( C_v, 3^K K^2 (C_c + 1) n_{\max} C_A^K \right). \tag{193}$$

*If instead $A$ satisfies Assumption 4.B then the bound becomes:*

$$\|\widetilde{\Phi}(h, A_n, f) - \Phi(h, A, \tilde{f})\|_2 \leq n^{-1}(3^K K n_{\max} C_A^K)^L \max\left( C_v, 3^K K^2 (C_v + 1) n_{\max} C_A^K \right). \tag{194}$$

*Proof.* Solve the recurrent in Lemma 14. $\qquad\square$

We now prove a more general version of Theorem 3

**Theorem 6.** *Let $A \in \mathcal{F}$ satisfying Assumption 2 with constant $C_A$ and Assumption 5.A with constant $C_c$ and resolution set $\mathcal{N} \subseteq \mathbb{N}$. Let $n \in \mathcal{N}$ and form the discretization $A_n$ as per Theorem 2. Let $h$ be normalized such that $|h| \leq 1$ element-wise and form the graphop neural network $\Phi(h, A, \cdot) : \mathcal{F} \to \mathcal{F}$ and $\Phi(h, A_n, \cdot) : \mathcal{F}_n \to \mathcal{F}_n$. We have the following approximation bound:*

$$d_M(\Phi(h, A, \cdot), \Phi(h, A_n, \cdot)) \leq n^{-1/2} P_1 \sqrt{C_A C_v + C_c n_{\max} C_A^K \cdot P_2} + C_v n^{-1} \tag{195}$$

*where $P_1 = 3^{KL}$ and $P_2 = 3^K K^2$.*

*If instead $A$ satisfies Assumption 5.B with constant $C_c$ then the bound becomes:*

$$d_M(\Phi(h, A, \cdot), \Phi(h, A_n, \cdot)) \leq n^{-1/2} P_1 \sqrt{\overline{C_A} C_v} + (C_c + 1) n_{\max} C_A^K \cdot P_2 + (C_v + 1) n^{-1} \quad (196)$$

*If instead $A$ satisfies Assumption 4.B then the bound becomes:*

$$d_M(\Phi(h, A, \cdot), \Phi(h, A_n, \cdot)) \leq n^{-1/2} P_1 \sqrt{\overline{C_A} C_v} + (C_v + 1) n^{-1} \quad (197)$$

*Proof.* The proof structure is similar to the proof of Theorem 2. For brevity of exposition, we will write $h$ as the (shared) set of parameters in the graphop neural networks and shorten $\Phi := \Phi(h, A, \cdot)$ and $\Phi_n := \Phi(h, A_n, \cdot)$. Fix $k \in \mathbb{N}$, we will bound $\sup_{\eta \in \mathcal{S}_{k,C_v}(\Phi)} \inf_{\overline{\eta}_n \in \mathcal{S}_{k,C_v}(\Phi_n)} d_{LP}(\overline{\eta}, \overline{\eta}_n)$.

Fix arbitrary $\overline{\eta} \in \mathcal{S}_{k,C_v}(\Phi)$. By definition of a $(k, C_v)$-profile, there exists $f_1, \ldots, f_k \in L^\infty_{\text{reg}(C_v)}([0,1])$ such that $\overline{\eta} = \mathcal{D}_\Phi(f_1, \ldots, f_k)$. For all $j \in [k]$, let $\mathcal{F}_n \ni f'_j : [n]/n \ni u \mapsto \int_{u-1/n}^u f_j(z) d\lambda(z)$. That $f'_j \in L^\infty_{\text{reg}(C_v)}([n]/n)$ is shown in Lemma 10. Set $\overline{\eta}_n = \mathcal{D}_{\Phi_n}(f'_1, \ldots, f'_k)$.

**Bounding $d_{LP}(\overline{\eta}, \overline{\eta}_n)$.** Fix some $\epsilon > 0$ to be specified later, by definition of $d_{LP}$, we need to bound, for each $U \in \mathcal{B}_k$:

$$\overline{\eta}(U) - \overline{\eta}_n(U^\epsilon) = \int_0^1 \mathbb{1}_{(f_1(x), \ldots, \Phi f_k(x)) \in U} d\lambda(x) - \sum_{u \in [n]/n} \frac{1}{n} \mathbb{1}_{(f'_1(u), \ldots, \Phi_n f'_k(u)) \in U^\epsilon}. \quad (198)$$

Notice that since $\lambda$ is the Lebesgue measure, one can denote $y_x = \lceil xn \rceil / n$ and write:

$$\overline{\eta}(U) - \overline{\eta}_n(U^\epsilon) = \int_0^1 \mathbb{1}_{(f_1(x), \ldots, \Phi f_k(x)) \in U} - \mathbb{1}_{(f'_1(y_x), \ldots, \Phi_n f'_k(y_x)) \in U^\epsilon} d\lambda(x). \quad (199)$$

Since the integrand is only positive when $(f_1(x), \ldots, \Phi f_k(x)) \in U$ and $(f'_1(y_x), \ldots, \Phi_n f'_k(y_x)) \notin U^\epsilon$ and this conjuction only happens when $\|(f_1(x), \ldots, \Phi f_k(x)) - (f'_1(y_x), \ldots, \Phi_n f'_k(y_x))\|_2 > \epsilon$, we have:

$$\overline{\eta}(U) - \overline{\eta}_n(U^\epsilon) \leq \lambda(\mathcal{E}(\epsilon)) \leq \sum_{j=1}^k \lambda(\mathcal{E}_j^0(\epsilon)) + \lambda(\mathcal{E}_j^1(\epsilon)), \quad (200)$$

where

$$\mathcal{E}(\epsilon) := \{x : \|(f_1(x), \ldots, \Phi f_k(x)) - (f'_1(y_x), \ldots, \Phi_n f'_k(y_x))\|_2 > \epsilon\}, \quad (201)$$

$$\mathcal{E}_j^z(\epsilon) = \{x : |\Phi^z f_j(x) - \Phi_n^z f'_j(y_x)| > \frac{\epsilon}{\sqrt{2k}}\}, \text{ for } z \in \{0,1\}, j \in [k], \quad (202)$$

where $B^0$ is the identity operator for any operator $B$ in the appropriate space.

**Bounding $\lambda(\mathcal{E}_j^1(\epsilon))$.** Fix $j \in [k]$, we have:

$$\int_0^1 |\Phi f_j(x) - \Phi_n f'_j(y_x)| dx \leq \sqrt{\int_0^1 \left(\Phi f_j(x) - \Phi_n f'_j(y_x)\right)^2 dx} \quad (203)$$

$$= \|\Phi f_j - \widetilde{\Phi_n f'_j}\|_{L^2} \quad (204)$$

$$\leq n^{-1}(3^K K n_{\max} C_A^K)^L \max\left(C_v, 3^K K^2 C_c n_{\max} C_A^K\right). \quad (205)$$

where the last line uses Lemma 15 under Assumption 5.A with constant $C_c$. Similar results are obtained for the other assumptions.

We also have:

$$\int_0^1 |\Phi f_j(x) - \Phi_n f'_j(y_x)| dx \geq \lambda(\mathcal{E}_j^1(\epsilon)) \cdot \frac{\epsilon}{\sqrt{2k}} + 0. \quad (206)$$

Thus selecting

$$\epsilon > \sqrt{2k\sqrt{2k} \cdot n^{-1}(3^K K n_{\max} C_A^K)^L \max\left(C_v, 3^K K^2 C_c n_{\max} C_A^K\right)} \quad (207)$$

makes $\lambda(\mathcal{E}_j^1(\epsilon)) < \frac{\epsilon}{2k}$. $\qquad \square$

**Bounding** $\lambda(\mathcal{E}_j^0(\epsilon))$**.** This step simply uses Lipschitzness of the graph signal to bound its discretization and is thus identical to that in Theorem 2. In short, choosing $\epsilon > \sqrt{2k}C_v/n$ gives $\lambda(\mathcal{E}_j^0(\epsilon)) = 0$.

**Bounding** $\sup_{\eta_n \in \mathcal{S}_{k,C_v}(\Phi_n)} \inf_{\eta \in \mathcal{S}_{k,C_v}(\Phi)} d_{LP}(\eta, \eta_n)$**.** This direction uses the same technique as in Theorem 2.

**Putting everything together** Defining $\overline{C_A} = (n_{\max} K C_A^K)^L$ gives a choice of:

$$\bar{\epsilon} = n^{-1/2} P_1 \sqrt{\overline{C_A} C_v + C_c n_{\max} C_A^K \cdot P_2} + C_v n^{-1} \tag{208}$$

where $P_1 = 3^{KL}$ and $P_2 = 3^K K^2$.

For other choices of assumptions, we obtain similar bounds.

## E Additional results

### E.1 Proof of Theorem 4

Because of completeness of $d_H$ in the space of closed subsets of $\mathcal{P}(\mathbb{R}^{2k})$ for every $k \in \mathbb{N}$, the statement in Theorem 4 is equivalent to showing that for each $k \in \mathbb{N}$, we have $\mathcal{S}_{k,L(n)}(A_n)$ converges to $\mathcal{S}_k(A)$ in $d_H$. We do this via a mollification argument.

**Definition 5** (Lipschitz mollifier)**.** *A Lipschitz mollifier in $\mathbb{R}$ is a smooth (infinitely differentiable) function $\phi : \mathbb{R} \to \mathbb{R}$ satisfying:*

1. *$\int_{\mathbb{R}} \phi(x)\mathrm{d}\lambda(x) = 1$.*

2. *$\lim_{\epsilon \to 0} \phi_\epsilon(x) := \lim_\epsilon \epsilon^{-1}\phi(x/\epsilon) = \delta(x)$ - the Dirac function.*

3. *Although not standard, we require $\phi$ to be $1$-Lipschitz and symmetric around $0$ ($\phi(x) = \phi(-x)$).*

*Given a measureable function $f \in L^\infty_{[-1,1]}(\mathbb{R})$, defines the convolution operation:*

$$f * \phi : x \to \int_{\mathbb{R}} f(y)\phi(y - x)\mathrm{d}\lambda(y). \tag{209}$$

The next result shows the existence of such a function:

**Lemma 16.** *Let $\phi : \mathbb{R} \to \mathbb{R}$ be:*

$$\phi(x) = \begin{cases} e^{-(1-x^2)^2}/Z & \text{if } |x| \leq 1 \\ 0 & \text{otherwise,} \end{cases} \tag{210}$$

*where $Z$ is a normalization constant to make sure that $\int_{\mathbb{R}} \phi \mathrm{d}\lambda = 1$. Then $\phi$ is a Lipschitz mollifier.*

*Proof.* The first property is built into the definition. The second property is obvious since the support of $\phi(x/\epsilon)$ is $[-\epsilon, \epsilon]$ and thus it converges to the Dirac function as $\epsilon$ goes to 0. Lipschitz-ness can be seen by computing the first derivative (since the function is smooth) over $(-1, 1)$ and see that it is bounded in $[-1, 1]$. Symmetry is also obvious since the function depends only on the absolute value of its argument. $\square$

Consequences of a Lipschitz mollifier include:

**Lemma 17.** *Let $\phi$ be a Lipschitz mollifier and $\epsilon > 0$, then for any measurable $f \in \mathcal{F}^\infty_{[-1,1]}(\mathbb{R})$, $f * \phi_\epsilon$ is $\max(1, \epsilon^{-2})$-Lipschitz and $\lim_{\epsilon \to 0} \|f - f * \phi_\epsilon\|_2 = 0$.*

*Proof.* For the first part of the statement, consider:

$$|f * \phi_\epsilon(x) - f * \phi_\epsilon(y)| = \left| \int_\mathbb{R} f(z)\phi_\epsilon(z-x) - f(z)\phi(z-y)\mathrm{d}\lambda(z) \right| \tag{211}$$

$$\leq \int_\mathbb{R} \left| \frac{f(z)}{\epsilon} \left( \phi\left(\frac{z-x}{\epsilon}\right) - \phi\left(\frac{z-y}{\epsilon}\right) \right) \right| \mathrm{d}\lambda(z) \tag{212}$$

$$\leq \frac{1}{\epsilon} \int_\mathbb{R} \left| \frac{z-x}{\epsilon} - \frac{z-y}{\epsilon} \right| \mathrm{d}\lambda(z) \tag{213}$$

$$\leq \frac{|x-y|}{\epsilon^2} \tag{214}$$

For the second part, we have:

$$|f * \phi_\epsilon(x) - f(x)| \leq \int_\mathbb{R} \phi_\epsilon(z-x)|f(z) - f(x)|\mathrm{d}\lambda(z) \tag{215}$$

$$\leq \int_\mathbb{R} \phi_\epsilon(z)|f(z+x) - f(x)|\mathrm{d}\lambda(z), \tag{216}$$

where we use two changes of variables.

Square both sides and take integral over $x$ and use Jensen inequality to get:

$$\|f * \phi_\epsilon - f\|_2^2 \leq \int_\mathbb{R} \int_\mathbb{R} (\phi_\epsilon(z))^2 (f(z+x) - f(x))^2 \mathrm{d}\lambda(z)\mathrm{d}\lambda(x). \tag{217}$$

Apply Fubini-Tonelli theorem to factorize the mollifier, we get:

$$\|f * \phi_\epsilon - f\|_2^2 \leq \int_\mathbb{R} \int_\mathbb{R} (f(z+x) - f(x))^2 \mathrm{d}\lambda(x)(\phi_\epsilon(z))^2 \mathrm{d}\lambda(z) \tag{218}$$

$$= \int_\mathbb{R} \left( \epsilon^{-1} \int_\mathbb{R} (f(z+x) - f(x))^2 \mathrm{d}\lambda(x) \right) (\phi(z\epsilon^{-1}))^2 \mathrm{d}\lambda(z). \tag{219}$$

Apply a final change of variable:

$$\|f * \phi_\epsilon - f\|_2^2 \leq \int_\mathbb{R} \left( \int_\mathbb{R} (f(z\epsilon+x) - f(x))^2 \mathrm{d}\lambda(x) \right) (\phi(z))^2 \mathrm{d}\lambda(z). \tag{220}$$

Therefore, as $\epsilon$ goes to 0, the inner integrand goes to 0. Since $f$ and $f * \phi_\epsilon$ are all bounded (as $f \in [-1, 1]$) for small enough $\epsilon$, we can apply dominated convergence to conclude that the integral itself goes to 0 and thus the 2-norm on the left hand side also goes to 0. $\square$

We are now ready to proceed with the proof of Theorem 4. Given an element $\eta \in \mathcal{S}_k(A)$, there exists a set of functions $F = \{f_1, \ldots, f_k\} \subseteq L^\infty_{[-1,1]}([0,1])$. Each of these functions can be extended to $\mathbb{R}$ by setting $f(x) = f(0)$ if $x \leq 0$ and $f(x) = f(1)$ if $x > 1$. Call the extended function $f'$. Now we can apply mollification convolution to each of them to get a family of functions $\{f_{j,\epsilon} := f'_j * \phi_\epsilon\}_{j\in[k],\epsilon>0}$. Recall that we have shown $f_{j,n} := f_{j,1/\sqrt{L(n)}}$ to be $L(n)$-Lipschitz for each $n \in \mathbb{N}$. Let $f''_{j,n}$ be the restriction of $f_{j,n}$ to $\mathcal{F}_n$. Then $f''_{j,n}$ is still $L(n)$-Lipschitz (in the metric on $1/n[n]$ induced by metric on $[0,1]$) and thus we can find a profile for $F''_n := \{f''_{1,n}, \ldots, f''_{k,n}\}$ in $\mathcal{S}_{k,L(n)}(A_n)$.

Furthermore, by property of mollifier and the fact that $L(n) \to \infty$ as $n$ goes to infinity, we have $\|f_{j,n} - f'_j\|_2 \to 0$ with $n \to \infty$. Since $f'_j$ is constant outside $[0,1]$, we also have $\|f_{j,n,|[0,1]} - f_j\|_\mathcal{F} \to 0$ with $n$. Using the same proof technique as Theorem 2, we can conclude that:

$$d_{LP}(\mathcal{D}_{A_n}(F''_n), \mathcal{D}_A(F)) \xrightarrow{n\to\infty} 0, \tag{221}$$

and thus $\mathcal{S}_k(A) \subseteq \lim_n \overline{\mathcal{S}_{k,L(n)}(A_n)}$. For the other direction, recall that $\overline{\mathcal{S}_k(A)} = \lim_n \overline{\mathcal{S}_k(A_n)}$ and that $\mathcal{S}_{k,L(n)}(A_n) \subseteq \mathcal{S}_k(A_n)$. Together with completeness of $d_H$, we conclude that

$$\lim_n d_H(\mathcal{S}_k(A), \mathcal{S}_{k,L(n)}(A_n)) = 0.$$

**Conjecture 1** (Action convergence of graphop neural networks). *Let $(A_n)_{n\in\mathbb{N}}$ be an action convergent sequence of graphops. Then $(\Phi(h, A_n, \cdot))_{n\in\mathbb{N}}$ is an action convergent sequence of P-operators.*

