# OpenReview forum: "Limits, approximation and size transferability for GNNs on sparse graphs via graphops"
_NeurIPS.cc/2023/Conference — NeurIPS 2023 poster_

### Official Review · Reviewer_3WrU · 2023-06-30

**Soundness:** 3 good
**Presentation:** 2 fair
**Contribution:** 3 good
**Rating:** 6
**Confidence:** 3

**Summary:**

This paper analyzes the transferability and approximation qualities of Graph Neural Networks (GNNs) when applied to graphs or Graph Signal Operators (GSOs) sampled from graph limit objects, known as graphops. The authors demonstrate that when a sequence of graphs is sampled from a graphop, the sequence converges according to a well-known metric. The paper also introduces the concept of graphop neural networks, which are utilized as a tool for analysis. It is shown that when a GNN is applied to a sequence of graphs sampled from a graphop, the output converges to the corresponding output from the graphop neural network applied to the graphop itself.

**Strengths:**

- Presents a novel integration of known techniques, specifically graphops and Graph Neural Networks (GNNs). The authors notably generalized the results for graphs converging to a graphon, which has been proven in [1,2,3].
- This research could potentially pave the way for further exploration into the transferability of GNNs in relation to sparse graphs. However, the authors miss to mention some work from Keriven et al. and Ruiz et al. which already consider non-dense graphs.

**Weaknesses:**

- The quality of writing and presenting is poor leading to bad comprehensibility. None of the theorems are self-contained. While the proofs seem correct, it is difficult to follow them due to notational issues and gaps in the argumentation (see below for a non-complete list). Note that I did not check all proofs thoroughly.
- The authors make some, at least, weird assumptions such as graphops being self-adjoint, but allowing them to be non-linear. The set of such operators is empty. Also, the choices of the convergence notion are not clear, even though a subsection is dedicated to it.
- Some assumptions appear restrictive, e.g.: The graphops are assumed to be sampled from $[0,1]$, in contrast to many other transferability papers (e.g. [2-4]). Assumption 3.A  also appears restrictive.
- The paper lacks clear organization and writing, with numerous notational errors scattered throughout. The order of presenting assumptions before other information also appears confusing. The appendix also contains many, at least, notational issues.
- The practical and theoretical implications of the results remain unclear. For instance, it would be beneficial to understand:
a) The potential impact of this work on theorists and practitioners, b) Examples of graphs for which the results are applicable, c) How can the convergence stated in Theorem 2 and 3 be experimentally demonstrated?, and d) The restrictiveness of the assumptions made in the study.


**Questions:**

- Could you clarify how your convergence definition aligns with the one proposed by Backhaus and Szegedy? In particular, how can we assure that convergence results still hold if we consider a sequence of Lipschitz constant $(C_{v_n})_n$ that grows sublinearly in n? Why does $d_M$ from Backhaus and Szegedy also converge to zero in this setting?
If this is sufficient, why do the authors change the notion of convergence?
What motivates the chosen notion of convergence? If not for mathematical convenience, what other reasons justify this choice?
- What implications do these results have for practitioners? Maybe the authors can provide some (simple) experiments, where Theorem 2 and 3 are verified? This would help for clarification and realizing an example (other than dense graph convergence which was already covered in previous works)
- What implications may these results have for developing new theories? For example, do you expect to achieve generalization bounds for sparse graphs such as in [4]?
- On line 73, could you explain why nonlinear operators are significant? My understanding is that all Graph Signal Operators (GSOs) are linear operators.
- You sample nodes deterministically, can you compare the loss of generality with works that do not assume this? E.g., [2,3,4]
- You mention that all graphops are self-adjoint. In light of this, why are nonlinear operators brought up? Note that every self-adjoint operator is linear.
- Regarding Theorem 4, could you clarify its utility? Also, the proof seems peculiar and certain terms such as $A$ and $A_n$ lack clear definitions.
For example, why does $S_{k,L(n)} (A_n) \subset S_k(A)$ hold? These profiles consist of different distributions, right? More details are provided below.

### Minor Issues and Questions:
- Line 12: Could you clarify why graphs sampled from the same graphop share structural properties? Especially, for graphops that are not induced by graphons.
- Line 24: Please consider adding [2,3].
- Line 43: Please refer to Keriven's work on "relatively sparse" graphs in the context of random graph models.
- Line 53: Do you mean "limits of GNNs of infinite sequences of graphs"?
- Line 58: [3] generalized the results to unbounded graphon operators and more diverse GNNs.
- Line 89: Ruiz's work also indicates a $\sqrt(n)$ dependence which decays to zero, this statement seems incorrect. Also, there are other works by Ruiz considering sparse graphs.
- Line 100: Random Graph Models (RGMs) essentially consist of graphons/kernels which are graph limits, and they are not opposed to these.
- Line 102: Could you define "closeness"?
- Line 119: Please mention that the entire work considers only spectral (polynomial) GNNs.
- Line 130: The space also contains limits of sparse graphs.
- Line 185: What does "positiveness" mean?
- Line 235: Do the results generalize to higher dimensional features?
- Line 246: What is $F_m$?
- Line 310: How do you define d_M for these non-linear operators that can neither be self-adjoint nor positive?
- Line 334: What do you mean with "wild"?
- Line 335: Could you elaborate on what "specifically designed for our discretization scheme" means?
- Line 341: Why "high probability"?
- Line 379: Could you clarify what this sentence means: "Then, A_n converges to the same limit as action convergence"?

#### Proof of Theorem 2:
- In bounding d_H, why is the maximum over certain quantities not considered?
- Line 731: Could you define the conditions for epsilon in the phrase "for some epsilon"?
There seem to be numerous typos throughout, such as in equation 56 and line 727, $S_k,C_v$.
- The definition of $U_\varepsilon$ is difficult to locate and appears incorrect.
Equality 60 should be an inequality.
- Line 763: What does $\bar{\mathcal{E}}$ denote?

#### Proof of Theorem 4:

- Line 1007: Could you clarify what it means for $f''$ to be Lipschitz in $l^2(n)$?
- Line 1011: What does the notation $\lim_\bar{S_{k,L(n)}(A_n)}$ signify?

[1] Ruiz, L., Chamon, L., & Ribeiro, A. (2020). Graphon neural networks and the transferability of graph neural networks. Advances in Neural Information Processing Systems, 33, 1702-1712.

[2] Keriven, N., Bietti, A., & Vaiter, S. (2020). Convergence and stability of graph convolutional networks on large random graphs. Advances in Neural Information Processing Systems, 33, 21512-21523.

[3] Maskey, S., Levie, R., & Kutyniok, G. (2023). Transferability of graph neural networks: an extended graphon approach. Applied and Computational Harmonic Analysis, 63, 48-83.

[4] Maskey, S., Levie, R., Lee, Y., & Kutyniok, G. (2022). Generalization analysis of message passing neural networks on large random graphs. In Advances in Neural Information Processing Systems.


**Limitations:**

The authors do not address limitations.

---

> ### Author Rebuttal · Authors · 2023-08-10
>
> We thank the reviewer for the feedback.
>
> We name the 5 points raised in the Weakness section W1-W5, points raised in Question Q1-Q7, and points raised in Minor Issues M1-M24. We write B-S to abbreviate Backhaus and Szegedy’s original paper.
> 1. For comments regarding the assumptions (W3), future implications (W5, Q2, Q3) and M1, M7, see the reply to all reviewers.
> 2. Typos, missing references, phrasing and notational errors (W1, W4, Q7, M2-7, M9, M17-18, M20-22, M24): We will correct them and highlight other elusive notations in the revision of the paper. We will also work on improving the clarity of the paper.
> 3. Self-adjoint and nonlinear operators (W2, Q4, Q6, M14): Indeed there are no self-adjoint (strictly)-nonlinear operators. However, we defined ‘nonlinear P-operators’ as those that are ‘not necessarily linear’ (line 180), including linear operators. This is important since a GNN layer itself (as opposed to just the GSO, which is linear) is a nonlinear operator and we would like to include it in the definition of P-operators. In fact, the whole (equivariant) GNN is also a nonlinear operator. This allows us to write Theorem 3, which uses $d_M$ - a metric originally derived to compare P-operators, to compare 2 GNNs. We will highlight this in the paper. Moreover, we only used the self-adjoint property of graphops to ensure that discretizing a graphop indeed gives a finite graph (Lemma 1). In Theorem 2 and 3, we no longer assume self-adjoint properties so at no point are there operators that are both (strictly)-nonlinear and self-adjoint.
> 4. Restrictive assumptions (W3): While the domain of the signal is [0,1], we can extend our results to any bounded domain, but currently we do not know of a proof for unbounded signal domains.
> 5. Comparison of the modified convergence and B-S original convergence (Q1, Q4): The proof of Theorem 3 with a slow-growing function of n mirrors the current proof verbatim. To gain intuition on the equivalence of the two convergence, note that the only difference between the two convergences is our use of $(k, C_v(n))$-profiles, which consider all bounded, $C_v(n)$-Lipschitz signals, instead of just $k$-profiles, which consider all bounded measurable signals. By letting $C_v(n)$ grow to infinity, we approximate all Lipschitz functions, which are dense in $L^2$ functions. The benefit of recovering B-S’s notion of convergence is that we inherit its useful properties, such as completeness of $d_M$ for linear operators. This modification allows us to use Lipschitzness of signals in our proofs of Theorem 2 and 3. This is necessary, following the insight: approximating functions by step functions on intervals of equal size (a discretization) requires a Lipschitz condition (consider extremely spiky functions, otherwise). This smoothness assumption is also ubiquitous in graphons and random graph kernels literature and may even be relaxed using our mollification argument (Theorem 4).
> 6. Proof of Theorem 4 (Q7): We will update a self-contained proof of Theorem 4 and correct the missing notations. Here, $A_n$ is the sequence of P-operators introduced in the proposition and A is the B-S original limit of $A_n$. A exists due to Cauchy-ness of $A_n$ in our metric implying Cauchy-ness in B-S’s original metric (as $(k,C_v(n))$-profiles grow to approximate $k$-profiles) and completeness of the space. In line 1011, $S_{k, L(n)}(A_n) \subset S_k(A)$ should have read $S_{k, L(n)}(A_n) \subset S_k(A_n)$, which holds since all signals considered in $S_{k, L(n)}$ are also considered in $S_k$, and concludes the other containment direction.
>
> Minor issues:
> - M6: We cited a wrong version of Ruiz et al. 2020: their IEEE TSP submission ‘Transferability properties of GNN’ has this $O(n^{-1})$ bound. It is unclear how sparse the graphs their new ‘sparse’ model ('GNN for Community Detection on Sparse Graphs') generates but personal communications and our current inspection suggests that their model is close to graphons. We will discuss this in the paper.
> - M8: We are referring to their Lipschitz assumption, which directly implies good approximation of a continuous GCN signal by a discrete sampled graph signal.
> - M10: Going by our definition of sparse graphs, graphon’s cut metric cannot distinguish them from the 0 graphon and thus only contains trivial limits (the 0 graphon) of such sparse graph sequence.
> - M11: Positiveness refers to positive-definite operators where $\langle Ax,x \rangle > 0$ for all $x$ in Dom(A).
> - M12: We do not see any immediate obstacle to this generalization.
> - M13: See line 109.
> - M14: $d_M$ only relies on $(k,C_v)$-profiles. For nonlinear operators one can still couple each signal with its image under the operator to define the profiles.
> - M15: The **image** of the operator is only required to be measurable and can be very discontinuous.
> - M16: Assumption 3A/B come with a resolution set that contains the possible sizes of our discretization. Outside of this set, the assumptions do not need to hold.
> - M17: ‘with high probability’ should be deleted.
> - M18: This means that 1. The sequence converges in both the B-S notion and our notion; and 2. The limits are the same.
> - M19: See Line 726 and 808. One can prove $\max_x F(x) \leq M$ by proving $F(x) < M$ for arbitrary $x$.
> - M20: We are bounding $d_{LP}$ directly via its definition (see line 494), and thus want to find an $\epsilon > 0$ such that the condition in the definition of $d_{LP}$ is satisfied.
> - M21: See Line 495.
> - M22: ${\bar{\mathcal{E}}}_{j,A}$ is a typo for ${{\mathcal{E}}^1}\_{j,A}$.
> - M23: $f’’$ is Lipschitz in the induced metric on the subspace $1/n [n] \subset [0,1]$ equipped with the Euclidean metric.
> - M24: The bar indicates closure of a set (of distributions) in the weak convergence sense and the limit is taken as n goes to infinity (the existence of such limit is due to existence of $\lim_n S_k(A_n)$ as shown in B-S and the fact that  $S_{k, L(n)}(A_n)$ converges to $S_k(A_n)$ set-wise).

---

> > ### Comment · Reviewer_3WrU · 2023-08-13
> >
> > Thank you for your clarifications and comprehensive response. In light of these, I've increased the scores for soundness and presentation by one point each, and the overall rating by two points. Both Table 1 and Figure 1 have been particularly helpful. However, I'd like to point out that the result from Keriven et al. [1] for relatively sparse graphs does not have a $n^{−1/2}$ convergence rate. Additionally, [4] presents results showing uniform convergence with respect to message passing networks, where the convergence rate depends on the dimension of the sample space. I do not assign a higher score due to my concerns about the presentation.

---

### Official Review · Reviewer_jR1r · 2023-07-01

**Soundness:** 3 good
**Presentation:** 3 good
**Contribution:** 3 good
**Rating:** 5
**Confidence:** 1

**Summary:**

This paper analyzes the transferability of Graph Neural Networks (GNNs) in terms of size. It proposes Graphop Neural Networks that operate on graphop signals and discretization of the network. It proves that the discretization of graphop and the graph are close, and consequently, two discretizations of different resolutions are close. It also shows that the corresponding Graphop Neural Network outputs are close which implies the transferability of GNN in size.

**Strengths:**

My background does not support me in evaluating the paper fairly. However, I believe the tackled problem of GNN size transferability is critical, and the author's analysis using graph limit and its discretization is promising. The assumption seems mild, which could make the theory applicable to most existing GNNs and real-world sparse graphs.

**Weaknesses:**

The related work session is scarce, which makes it more difficult for me to understand the background. The paper is purely theoretical, which is fine. Still, it will be more sensible for readers like me if the authors can provide some form of experiment on synthetic graphs of variable size to better understand the implication of the theorem.

**Questions:**

- What is the key difference between this work and the graphon neural network which also adopt a graph limit perspective?
- The bounds require the number of vertices in both graphs to be sufficiently large. Then how to interpret the bound when one graph is large and the other is smaller, which is more important for graph size transferability, and also two small graphs.


**Limitations:**

Review Summary:
Due to my lack of background knowledge, I could not appropriately relate it to previous research and assess the proof fairly. However, the paper addresses a crucial problem in graph learning and seems to provide a sound theoretical analysis, which leaves it on the borderline.

---

> ### Author Rebuttal · Authors · 2023-08-10
>
> We thank the reviewer for their feedback and address individual questions and concerns:
> 1. Scarcity of related work: Past works on size generalizability of GNNs rely either on graphons or random graph models induced by a kernel, while we focus on deterministic sparse graphs. In the latter regime, we only are aware of Roddenbury et al., 2021. This is not unexpected as many authors in the literature agree with us that the analysis of sparse and bounded-degree graphs is challenging and intricate (more in our reply to all reviewers). That said, we will include a more detailed comparison with results obtained from random graph models and extended graphon models in the final version of the paper.
> 2. Comparison to graphon neural network: Our work shares a lot of structure with Ruiz et al. 2020’s graphon neural network, especially in our use of graph limits. One major distinction (see table in our reply to all reviewers), is that the graphon limit of sparse and bounded degree graphs is the constant 0 graphon. This makes their approximation bound between a finite graph sampled from a graphon to the limiting graphon vacuous.
> 3. Requirement of a large number of vertices: our bounds are non-asymptotic, quantitative and have a specific rate of convergence, which is one of the strengths of our results.  This means that our results work for all finite graph sizes. In particular, our transferability bound between a graph of size m and n states that the transferability gap scales with order $(\min(m,n))^{-1}$.
>
> We will work on improving the accessibility of the writing in our revision. We would also love to help you understand more about our line of work, so please do not hesitate to post more questions that you think we can help you answer.

---

> > ### Comment · Reviewer_jR1r · 2023-08-12
> >
> > I thank the authors' response. I believe the paper still has strong theoretical merit, while a clear presentation and application to real-world datasets will clarify the impact of the paper. Hence, I am keeping my score.

---

### Official Review · Reviewer_E8cZ · 2023-07-05

**Soundness:** 3 good
**Presentation:** 2 fair
**Contribution:** 2 fair
**Rating:** 3
**Confidence:** 2

**Summary:**

This paper explores the theoretical perspective of whether graph neural networks (GNNs) can generalize to graphs that are different from the ones they were trained on. The authors study the transferability and approximation results via graph limits, including sparse graphs such as bounded-degree or power law graphs. The paper presents various notions of graph limits and develops quantitative bounds on the distance between a finite GNN and its limit on an infinite graph. The authors also verify the regularity assumption for various graph sequences in this study. Overall, the paper's contributions include a theoretical framework for studying the generalization of GNNs to different graphs, as well as insights into the regularity of graph sequences and the use of graph limits for approximation and transferability analysis.

**Strengths:**

S1. The paper takes a unique perspective of studying the transferability and approximation results of GNNs via graph limits. The proposed approach is novel and provides a theoretical framework for studying the generalization of GNNs to different graphs.
S2. The authors provide rigorous mathematical proofs and analysis to support their claims, which potentially have important implications for the practical use of GNNs in real-world applications.
S3. The paper's results hold for both dense and sparse graphs, and various notions of graph limits, which makes the paper's contributions applicable to a wide range of graph-based learning tasks.

**Weaknesses:**

W1. Lack of real-world application examples: The paper focuses primarily on the theoretical aspects of graph limits and their implications for GNN generalization. However, it would be valuable to include real-world application examples where the proposed approach can be applied and demonstrate its practical utility. This would provide concrete evidence of the significance and relevance of the work in real-world scenarios.
W2. Limited experimental evaluation: While the paper includes some experimental results, the evaluation could be more extensive and diverse. It would be beneficial to include experiments on a wider range of datasets and graph structures to demonstrate the effectiveness and generalizability of the proposed approach. Additionally, providing a detailed analysis of the experimental results, including statistical significance tests and comparisons with baseline methods, would further strengthen the empirical findings.
W3. Lack of comparison with existing methods: The paper could benefit from a more comprehensive comparison with existing methods that address the generalization of GNNs to different graphs. This would provide a clearer understanding of the novelty and effectiveness of the proposed approach. Including a comparison with state-of-the-art methods and discussing the advantages and limitations of the proposed method in relation to existing approaches would strengthen the paper.

**Questions:**

Q1. Could you expand the experimental evaluation by including experiments on a wider range of datasets and graph structures? This would provide a more comprehensive understanding of the effectiveness and generalizability of your proposed approach. Additionally, providing a detailed analysis of the experimental results, including statistical significance tests and comparisons with baseline methods, would strengthen the empirical findings.
Q2. Can you provide real-world application examples where your proposed approach can be applied? Demonstrating the practical utility of your approach in real-world scenarios would further highlight the significance and relevance of your work.
Could you discuss the limitations of your proposed approach in more detail? Specifically, addressing the scalability of the approach to larger graphs, considering different types of GNN architectures, and exploring the impact of different graph properties on generalization performance would be interesting.
Q3. Have you considered the computational complexity of your proposed approach? It would be helpful to discuss the computational requirements and scalability of your method, especially when applied to larger graphs.
Q4. Can you provide more insights into the regularity assumption verified for various graph sequences? How does the regularity assumption impact the generalization performance of GNNs, and are there any specific conditions or properties that need to be satisfied for effective generalization?
Q5. Can you discuss potential future directions for your research? Specifically, addressing the scalability of the approach, exploring different types of graph structures, and considering the impact of different graph properties on generalization performance would be valuable areas for further investigation.

**Limitations:**

The paper does not explicitly address the potential negative social impact of the proposed approach. While the paper focuses primarily on the theoretical aspects of graph limits and their implications for GNN generalization, it would be beneficial to include a discussion on the potential ethical implications of the proposed approach. Specifically, the authors could consider the potential impact of their work on issues such as privacy, fairness, and bias in machine learning.

Additionally, the paper could benefit from a more comprehensive discussion of the limitations of the proposed approach. While the authors acknowledge some limitations, such as the need for regularity assumptions and the limited experimental evaluation, a more detailed discussion on the potential limitations and their impact on the generalization performance of GNNs would be valuable.

---

> ### Author Rebuttal · Authors · 2023-08-10
>
> We thank the reviewer for their feedback and would like to address the individual points.
> - W1. Real-world applications of GNNs on very sparse graphs (bounded degree, linear order of edges, etc.) have been demonstrated in various disciplines elsewhere. Our paper aims to explain the transferability observed widely in practice in a rigorous theoretical framework. Theory for these very sparse graphs is extremely limited in the literature and thus, lots of existing empirical findings for them are not accounted for rigorously. For example, all molecular graphs are bounded-degree, and thus extremely few existing theories can systematically explain transferability between molecule graphs even when they share lots of structural similarities (for example graphs of polymers with different numbers of units). Other realistic graphs, such as road networks, social networks, evolutionary trees, grammar trees are also mostly sparse due to physical constraints. These have also been studied extensively elsewhere and we refer the reviewer to applied publications for these phenomena.
> - W2. We believe the reviewer is mistaken since we do not include any experimental results in this paper, let alone strengthen them. Our main contribution is a theoretical analysis of transferability of GNNs, rather than a proposal of a new method or a new architecture. That being said, we will add proof-of-concept experiments to demonstrate our Theorems.
> - W3. We will extensively add to our related work section. See also more details in the response to all reviewers. However, since so few theoretical works in the literature share the same emphasis as ours (transferability on very sparse graphs, which are notoriously hard to analyze), it is unclear what the ‘state-of-the-art methods’ are.
> - Q1. Please refer to the reply to W2 above.
> - Q2. Relevance of our analysis to real-world applications: please refer to the response to W1. Scalability of our approach to larger graphs: since our bounds are quantitative and non-asymptotic, they apply to graphs of any size, large or small, and even graph limits. In this paper, we only consider spectral GNNs to demonstrate some main takeaways in terms of the metric and the corresponding convergence. However, we believe our approach works for other GNN models, such as graph isomorphism networks (GIN). We also agree that studying how different graph structures impact rate of convergence under the metric d_M is an interesting direction forward, but is outside the scope of this paper.
> - Q3. The main contribution of this paper is a convergence and transferability analysis of existing GNN models, as opposed to introducing new models. That said, we will provide small experiments demonstrating our Theorems in the final version of the paper.
> - Q4. For a discussion of our assumptions, please refer to the general address to all reviewers.
> - Q5. Please refer to the “Implication for future empirical and theoretical work” section in our reply to all reviewers.
> Limitations: we will address some ethical concerns of GNN developments. That said, seeing that our work theoretically verifies observed transferability experiments done elsewhere, we share the same ethical concerns with most papers in the GNN literature without meaningful deviations.

---

> > ### Comment · Reviewer_E8cZ · 2023-08-15
> > **Acknowledgement of rebuttal**
> >
> > I thank the authors for the replies. However, some of the replies only amplified my concerns.
> >
> > - W1. I understand the theoretical nature of this work, but it is still beneficial for even a very theoretical work to clearly discuss its application domains, to allow proper leverage of the theoretical findings. I see the authors mention molecular graphs, road networks, grammar trees etc. These are good application examples themselves, but it looks like the authors are rather reluctant to properly discuss them in the paper.
> > - W2. It is good to see that the authors have plans to add some proof-of-concept experiments, but with the current lack of details, I cannot tell what kind of experiments can be added and how helpful they would be to support the theoretical findings.
> > - W3. "State-of-the-art" means anything that exists and can best solve the problem at hand. I hoped the authors could clearly discuss the status of existing studies and enhance the discussions in the paper, instead of saying "it is unclear what the state-of-the-art methods are". A statement like this is not appropriate in the rebuttal, and certainly even more inappropriate to appear in a paper.
> >
> > Due to these amplified concerns, I had to reduce my rating.

---

> > > ### Author Response · Authors · 2023-08-16
> > > **Regarding "Acknowledgement of rebuttal"**
> > >
> > > We thank the reviewer for the second reply and will now address the extra concerns highlighted:
> > > - W1. We agree with the reviewer that it is important to highlight the applications that can benefit from our theoretical analysis. We will add a discussion of those to the paper.
> > > To the best of our knowledge, ours is the first paper to rigorously  prove transferability of sparse graph sequences. Therefore, all applications that involve size transferability of GNNs on graph sequences satisfying our assumptions (see Table 1 in the additional rebuttal pdf) are directly verified by our theorems. Furthermore, by using a framework general enough to include graph sequences of various sparsity, and by demonstrating that non-regular graph sequences too (e.g. polymer graphs, see Table 1 and Figure 1) can satisfy our assumptions, we argue that our theoretical findings can impact many future applications. These discussions will be present in the revision of the paper. We would like to reiterate that our examples of molecular graphs, road networks, grammar trees are all sparse, bounded-degree graphs, which are what our results cover, while most existing theoretical approaches do not cover them.
> > > - W2. We will verify the statement of our theorems by sampling graphs of various sizes from a limiting object, and then empirically checking the distances between the GNNs viewed as operators. We will also design robustness tests to understand the extent our results still hold or fail as our theoretical assumptions on the graph structures are violated. This will give us a further understanding about the results that is useful for practitioners.
> > > - W3. We believe there was a misunderstanding in our statement regarding comparisons to state-of-the-art methods that we hope to clarify. The current theoretical state-of-the-art is summarized in Table 1 and discussed in the “Related work” session of the main paper. Among the existing works, none gives quantitative bounds for multilayer GNNs while covering different graph sparsity regimes - which is the setting of our paper. For example, for bounded-degree graphs (first column of Table 1), only one other paper gives transferability guarantees for 1-layer GNN, and with an inexplicit bound. We will add this to the paper, along with a more detailed discussion and comparison to random graph frameworks (see “Deterministic vs random graph” in our reply to all reviewers). If the reviewer is asking for a comparison to state-of-the-art methods in applied GNN research: our paper currently does not suggest any modifications to these pipelines used by practitioners. Rather, we seek a deeper understanding of existing methods and datasets by providing an analysis of how they would perform as they are applied to graphs of a different size as the training data. Our analysis is complementing and explaining empirical analyses.

---

### Official Review · Reviewer_QqZq · 2023-07-10

**Soundness:** 3 good
**Presentation:** 3 good
**Contribution:** 2 fair
**Rating:** 6
**Confidence:** 3

**Summary:**

This work is essentially extending the existing theoretical study of GNNs through graphons by replacing them with graphops, an operator view on graphs that allows for results of sparse graphs. Then, standard results on size generalization for instance, are presented.

**Strengths:**

The paper is technically sound as far as I could check. The theoretical contributions are non-trivial and present enough mathematical rigor for this venue. It is a first step towards moving being graphon-restricted analysis of GNNs.

**Weaknesses:**

The issue I have with this paper is its usefulness for machine learning applications. We understand the space L^1([0,1]^2) pretty well for graphons, but the newly introduced operator view from Backhausz and Szegedy isn't clear to me yet. Could the authors give the readers more intuitions on how P-operators can generate real-world graphs? Remember that in the end of the day the GNN is applied to real-world graphs. Graphons can be seen as latent factor models, which is somehow interpretable as a graph generating process. How can P-operators be seen by the readers? This might be a flaw in my understanding that the authors can help me with, so I'm willing to raise my score. In particular how can  (k,L)-profiles be interpreted in real-world? Graphs as operators make sense in a lot of fields, but I'm a little concerned that graph data isn't fit to this view.

**Questions:**

Could the authors give the readers more intuitions on how P-operators can generate real-world graphs?
how can  (k,L)-profiles be interpreted in real-world?

---

> ### Author Rebuttal · Authors · 2023-08-10
>
> We thank the reviewer for their feedback. Regarding the questions and concerns:
> 1. On the view of graphs as P-operators: We first want to note that P-operators are very general operators satisfying a bounded (operator) norm condition. Viewing graphs as operators is not unfamiliar to finite graph analysis: the adjacency matrix of an n-vertex graph is a matrix and thus, a linear operator $A$ from $\mathbb{R}^n$ to $\mathbb{R}^n$, whose image describes a single layer of message passing algorithm: if $f \in \mathbb{R}^n$ is a vector of node features then $Af$ is the output of a single layer of message passing. The analysis of graphons W (dense graph limits) also makes use of properties of the Hilbert-Schmidt operator $H$ defined as $Hf(y) = \int_0^1 W(x,y)f(x) dx$. Similarly, the image of this operator is the pre-activation of a single layer of graphon neural networks. In this view, one can even bypass the use of a generation model (kernels/graphons) and directly model graphs by the images (actions) of their adjacency matrices or their transformations.
>
> 2. P-operators in real-world graphs: While dense graph limits such as graphons are amenable to more analysis tools, we would argue that P-operators as graph limits are more realistic for real-world graphs, albeit with a more difficult analysis - as you have seen in this paper. This is because real-world graphs are very rarely dense (at least quadratic order of edges) or even relatively sparse ($n\log n$ order of edges). Molecule graphs, evolutionary trees, grammar trees, social networks, for example, are usually sparse, bounded-degree even, because of real-world physical constraints. Thus, P-operators capture more realistic graphs in real-world datasets. Furthermore, graphon limit model is not enough: by definition, under the graphon model, the limit of these sparse graphs vanishes and cannot be distinguished from the constant 0 graphon, making their bounds trivial. It is also worth pointing out that both finite graphs and graphons are examples of graphops (which in turn are P-operators), so we have a larger graph limit model altogether.
>
> 3. $(k,L)$-profiles real-world interpretation: It is easier to gain intuition in real-world examples by considering $k$-profiles. After all, in Section 4.4 we show that with minimal overhead, $(k,L)$-profiles yield the same convergence as $k$-profiles. When $k = 1$, an element of a $1$-profile of a finite $n$-vertex graph is a joint probability distribution of the form $(X, Y)$ where $X$ is drawn from an empirical distribution supported on some node features $f$ and $Y$ is drawn from an empirical distribution supported on the image of $f$ after $1$ round of message passing on the graph. The $1$-profile is then the set of such distributions when $f$ is taken over a set of reasonably regular (measurable, bounded, Lipschitz) node features. Therefore, the $1$-profile couples the input and output of a message passing layer and captures their dependence structure across all reasonable node feature vectors. Please refer to Backhausz and Szegedy’s original paper for some illustrations of $k$-profiles. We will provide our own illustrations in the next revision of the paper.
>
> 4. Graph generation via P-operators: Please refer to the reply to all reviewers section.

---

> > ### Comment · Reviewer_QqZq · 2023-08-11
> > **ack rebuttal**
> >
> > Thank you for the reply. I think the paper still needs intuition on the relationship to real-world graph data, that was not provided in this reply. The paper still has valid theoretical contributions, so I keep my score.

---

### Author Rebuttal · Authors · 2023-08-10

We thank the reviewers for their feedback and would like to emphasize:
1. Our work establishes quantitative, non-asymptotic transferability bounds for a wide range of sparse graphs, which are notoriously hard to analyze. Please refer to Table 1 in the additional rebuttal pdf file for a concise comparison to related works. This will be in the final version of the paper.
2. Sparse graphs are notoriously difficult to analyze:
- Regularity assumptions are likely necessary: Although some of our assumptions (3A and 3B) seem restrictive, only one of them needs to hold for the results to go through. Furthermore, an unconditional proof of our Theorem 2 would solve the Aldous-Lyon conjecture for bounded-degree graph limits. While this does not prove such a result is impossible, we want to point out that graphops convergence is intricate and nontrivial to analyze.
- Pathological behavior of graph spectrum in the limit: It is well known that for many bounded-degree graph sequences, eigengaps do not converge and that eigenvalues and eigenvectors may not even exist for limiting operators.  Therefore, these sequences are not amenable to spectral methods that use eigenvalue/eigengap convergence. (e.g. Levie et. al, 2021)
3. Sparse graphs and bounded-degree graphs are more realistic in real-world models. See Table 1 for examples that satisfy our assumptions. We have recently verified that polymer graphs (refer to Figure 1 in the additional rebuttal pdf for an illustration of this graph), which model the repetitive structure of polymer molecules, satisfy our assumptions. Unlike other example graphs in the paper, polymer graphs are not necessarily regular graphs.
4. Deterministic vs random graph: at the time of writing, we considered these two distinct approaches to size transferability. While random graph models are more flexible and usually have stronger results and wider tools for analysis, results for random graph models usually only hold with high probability. In comparison, deterministic graph sampling is more suitable if there is already a deterministic sequence of graphs one has in mind (for example, images of certain resolutions) and wants to compute transferability bounds between them. Results for deterministic graph sequences must also hold in the worst case since there is no randomness to downplay the probability of sampling these graphs. However, seeing that the techniques in both approaches are similar in existing literature, in the final version of the paper, we will include a more thorough discussion of random graph models.
5. Experiments: Comprehensive empirical studies of transferability on sparse graphs have already been done extensively in the literature, especially since sparse graphs are more common in real-world datasets than dense ones. We will add proof-of-concept experiments that verify some of our results in the final version of the paper.
6. Implication for future empirical and theoretical work: Since our approach to analyzing very sparse graphs is very new, there are lots of exciting avenues to consider in the future, both in practice and in theory. By introducing P-operators, graphops, and the accompanying metric (with our relaxation to $(k, L)$-profiles for easier use), we contribute to the toolbox of techniques used to analyze and compare GNNs of different graph sizes. This may have impacts on other aspects of GNNs, such as expressiveness (universality theorems under this metric) or proving new generalization bounds beyond size transferability. As our results apply to a much wider range of graph sequences than previous ones, they may also serve as a unifying framework for future studies. From an empirical standpoint, our results ensure size transferability under appropriate structural assumptions (they are sampled from the same regular graphop) between two graphs. Testing how robust size transferability is to failure of such structural assumptions will help practitioners predict transferability and non-transferability of sparse graph sequences. Finally, the connection between sparse graph convergence and weak convergence of distributions is yet another rich future direction. These two concepts are tied together since the $d_M$ metric is built on top of Levy-Prokorov distance ($d_{LP}$), which metrizes weak convergence and most of the proofs boil down to bounding $d_{LP}$. This connection may allow techniques used in optimal transport or Wasserstein distance computation to benefit this line of work. More directly related to this work, generalizing our results to higher dimensional node features appears to be straightforward. Further relaxing Assumptions 3A and 3B is also interesting and would greatly improve this approach.
7. Graph generation via P-operators: The discretization scheme introduced for graphop can be roughly understood as partitioning the vertex set into finitely many sets and merging nodes and their connections in these sets, with an appropriate scaling of the edge weights. Imagine blurring an n by n matrix into an  n/2 x n/2 matrix by doing some form of average pooling over each distinct 2x2 square. This makes rigorous the notion of making a high-resolution graph on a huge number of vertices more ‘blurry’ by merging nodes in a way that still maintains some smoothness conditions of Assumptions 3A/3B, which is reminiscent of the real-world procedures of training with low-resolution images before fine-tuning with higher-resolution ones, or sampling from low-frequency graph Fourier transform domain. We will clarify these intuitions in the final version of the paper.

Finally, we note that all citations that appear in our rebuttals can be found in the reference section of the current version of the paper, except that Ruiz et. al, 2020 now refers to their IEEE TSP submission ‘Transferability properties of GNN’ available on arXiv.

---

### Comment · Area_Chair_46rT · 2023-08-13

Thanks to all reviewers and authors for their work on this submission.

As the discussion period starts, I want to make sure that reviewers have read the author's response.

This can be done either by communicating with authors, or in private conversation within the reviewing team.

---

### Decision · Program_Chairs · 2023-09-21

**Decision:**

Accept (poster)

**Comment:**

This paper is a study of transferability of GNN with graphops. It was overall well received by the reviewers except that the lack of real world examples was considered as an important issue of the paper. I believe there is some truth in this comment, but the theoretical contributions are sound, and are worthy of a NeurIPS publication.

Please incorporate the remaining reviewers' feedback for the camera-ready version.